# Recombinant expression systems for production of stabilised virus-like particles as next-generation polio vaccines

Lee Sherry [1,8], Mohammad W. Bahar [2,8], Claudine Porta[2,3,8], Helen Fox[4,8], Keith Grehan [1], Veronica Nasta[2,5], Helen M. E. Duyvesteyn[2], Luigi De Colibus[2], Johanna Marsian[6], Inga Murdoch[6], Daniel Ponndorf[6], Seong-Ryong Kim[6], Sachin Shah[6], Sarah Carlyle [4], Jessica J. Swanson[1], Sue Matthews[1], Clare Nicol[1], George P. Lomonossoff [6] ✉, Andrew J. Macadam [4] ✉, Elizabeth E. Fry [2] ✉, David I. Stuart [2,7] ✉, Nicola J. Stonehouse [1] ✉ & David J. Rowlands [1] ✉

Polioviruses have caused crippling disease in humans for centuries, prior to the successful development of vaccines in the mid-1900's, which dramatically reduced disease prevalence. Continued use of these vaccines, however, threatens ultimate disease eradication and achievement of a polio-free world. Virus-like particles (VLPs) that lack a viral genome represent a safer potential vaccine, although they require particle stabilization. Using our previously established genetic techniques to stabilize the structural capsid proteins, we demonstrate production of poliovirus VLPs of all three serotypes, from four different recombinant expression systems. We compare the antigenicity, thermostability and immunogenicity of these stabilized VLPs against the current inactivated polio vaccine, demonstrating equivalent or superior immunogenicity in female Wistar rats. Structural analyses of these recombinant VLPs provide a rational understanding of the stabilizing mutations and the role of potential excipients. Collectively, we have established these poliovirus stabilized VLPs as viable next-generation vaccine candidates for the future.

Poliovirus (PV), the causative agent of poliomyelitis, was a major public health concern during the 19th and 20th centuries, causing paralysis and death on a global scale, especially in children[1]. To combat this scourge, a vigorous vaccine development programme resulted in the Salk inactivated polio vaccine (IPV) and Sabin live-attenuated oral polio vaccine (OPV). The Global Polio Eradication Initiative, launched in 1988 using these vaccines, successfully reduced wild-type PV (wt PV) incidence by over 99%[2]. Of the three wt PV serotypes (1, 2 and 3), PV2 and PV3 have been eradicated[3,4], and currently wt PV1 remains endemic only in Afghanistan and Pakistan[2]. OPV is preferred in most regions due to its low cost, ease of administration and ability to induce comprehensive immunity in the gut, the primary site of PV infection and replication, thereby breaking virus transmission. IPV induces humoral immunity, preventing

[1]Astbury Centre for Structural Molecular Biology, School of Molecular and Cellular Biology, Faculty of Biological Sciences, University of Leeds, Leeds, UK. [2]Division of Structural Biology, University of Oxford, The Henry Wellcome Building for Genomic Medicine, Oxford, UK. [3]The Pirbright Institute, Surrey, UK. [4]Division of Vaccines, Medicines & Healthcare products Regulatory Agency (MHRA), Herts, UK. [5]Magnetic Resonance Center CERM, University of Florence, Sesto Fiorentino, Florence, Italy. [6]John Innes Centre, Norwich Research Park, Norwich, UK. [7]Diamond Light Source, Harwell Science and Innovation Campus, Didcot, UK. [8]These authors contributed equally: Lee Sherry, Mohammad W. Bahar, Claudine Porta, Helen Fox. ✉e-mail: george.lomonossoff@jic.ac.uk; andrew.macadam@mhra.gov.uk; elizabeth.fry@strubi.ox.ac.uk; dave.stuart@strubi.ox.ac.uk; n.j.stonehouse@leeds.ac.uk; d.j.rowlands@leeds.ac.uk

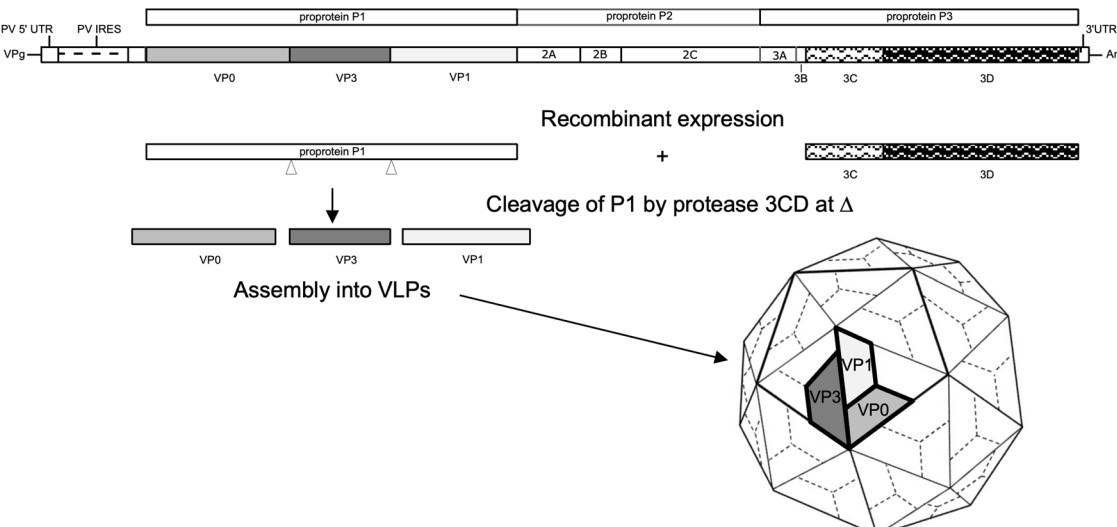

**Fig. 1 | Schematic of PV genome and VLP expression strategy.** The capsid region P1 and the viral protease, 3CD, were introduced into each of mammalian, yeast, insect and plant expression systems for production of PV VLPs.

viraemic spread and disease in the vaccinee but does not preclude infection and potential onward transmission[5,6].

Despite the efficacy of these vaccines, there are concerns associated with their continued use as we approach the endgame of eradication[1]. The attenuated strains used in OPV manufacture are genetically unstable and can regain neurovirulence, even causing rare cases of vaccine-associated paralytic poliomyelitis in OPV recipients or becoming a source of circulating vaccine-derived PV (cVDPV)[7]. The number of cVDPV cases of disease now exceeds that of wild PV cases and has increased through person-to-person transmission in areas with low vaccination coverage[2,8]. Additionally, novel PVs can emerge via genetic recombination with non-polio enteroviruses[9,10]. Immunodeficiency-associated vaccine-derived PV (iVDPV) in immune-compromised individuals also contributes to the risk of circulating viruses, since chronic virus infection can result in life-long virus shedding[11,12]. Recently, highly modified OPV strains have been developed, greatly reducing the potential for reversion to virulence and for recombination[13,14]. These strains are recommended for emergency use to control outbreaks. The alternative vaccine, IPV, requires the production of large quantities of infectious PV, posing significant risks of accidental release[15] and the formalin inactivation process leads to changes in the antigenic structure of the capsid[16]. In order to overcome the bio-safety concerns associated with current PV vaccines, there is a pressing need for alternatives that no longer rely on infectious virus cultivation, and which will be suitable for the post-eradication era.

PV, an enterovirus within the *Picornaviridae* family, is a positive-sense single-stranded RNA virus with a 7500 nucleotides genome enclosed within a non-enveloped icosahedral protein capsid of ~30 nm diameter[17,18]. The PV genome is translated from a major open reading frame into a polyprotein comprising three regions (P1, P2 and P3), which is proteolytically cleaved into the mature viral proteins (Fig. 1)[18]. The structural protein precursor P1 encodes the viral capsid proteins, while P2/P3 encode proteins involved in genome replication, polyprotein processing and modification of the host cell environment[19]. The viral protease precursor 3CD is responsible for cleaving P1[20,21], which is initially processed into capsid proteins VP0, VP1 and VP3. Encapsidation of the viral RNA to form the mature virion is associated with autocatalytic cleavage of VP0 into VP2 and VP4, with a concomitant increase in particle stability[22,23]. Mature virions comprise 60 copies of the VP1–VP4 protomer and a single copy of the viral genome. Naturally occurring empty capsids (ECs) also form

during PV replication and, in the absence of genome, their VP0 proteins remain uncleaved[24]. PV particles display two distinct antigenic structures, the native D antigen (D Ag) associated with mature infectious virus and the non-native C antigen (C Ag), characteristic of non-infectious particles[25,26]. The D Ag elicits a protective immune response but can be converted to the C Ag, for example by heating[27]. Conversion of D Ag to C Ag is associated with the expansion of the particle by approximately 3%, resulting in the loss of the ability to induce protective immunity, making this unsuitable as a vaccine[27,28]. It has been proposed that naturally occurring ECs could be useful as non-replicating virus-like particle (VLP) vaccines against PV[29]. Although ECs are inherently unstable outside of cells, rapidly converting from D Ag to C Ag[30], genetic manipulation of the capsid protein sequences has resulted in the generation of virions and ECs that are stabilised in the immunogenic D Ag conformation[31].

VLP vaccines against hepatitis B[32] and human papillomavirus[33] are widely used and have set a precedent for the efficacy of this technology. VLPs mimic the repetitive structure of native viral particles which renders them highly immunogenic but lack the viral genome making them safer and potentially cheaper vaccine candidates[34,35]. Recombinant PV VLPs have been produced in yeast, insect, plant and mammalian cells[29,36–41] by expressing the structural protein precursor P1 together with the 3CD protease precursor, which is sufficient for processing P1 into the VP0, VP3 and VP1 capsid protein subunits.

By incorporation of the mutations identified in viruses, we previously demonstrated production of PV recombinant stabilised VLPs (rsVLPs) in yeast, plant and mammalian cells[38,39,42]; here, we present a comparative analysis of PV rsVLPs produced in these three cell types and in insect cells using baculovirus-mediated expression. We compare the levels of native D Ag produced in these four systems and utilise cryogenic electron microscopy (cryoEM) to compare the structures of the rsVLPs to those of wt PV. We demonstrate that rsVLPs are protective using a transgenic mouse challenge model, before assessing their ability to induce a neutralising antibody response in Wistar rats, the current batch release test for licensed inactivated PV vaccines. We show that in the presence of adjuvant, our rsVLPs outperform the current IPV, with the potential for dose-sparing. This work establishes PV rsVLPs as a source of virus-free vaccine for the post-eradication era.

## Results

### Identification of stabilising mutations and design of stable VLPs for PV serotypes 1–3

Although empty PV capsids have been shown to be inherently unstable outside the confines of the cell[30], our previous work identified candidate mutations that stabilise temperature-sensitive PV mutants when grown at semi-permissive temperatures[31,43]. Importantly, none of these stabilising mutations were in known antigenic regions for any of the three serotypes (PV1 Mahoney, PV2 MEF-1 and PV3 Saukett), and therefore were unlikely to affect antigenicity[31]. These mutations were incorporated and tested in multiple configurations for each serotype to determine the best-performing constructs in terms of capsid assembly and thermostability for expression in recombinant systems. The selected stable capsid (SC) constructs were as follows: PV1-SC6b, PV2-SC6b and PV3-SC8. The corresponding sets of amino acid substitutions and their location within the capsid are described in Fig. 2a, b. They are clustered at pentamer and protomer interfaces and below the receptor binding site, such that they are not prominent on the VLP outer surface and are generally distinct from identified antigenic sites (Fig. 2c).

### Comparison of selected recombinant expression systems

The successful development of novel PV VLP-based vaccines is dependent on them fulfilling several important criteria. They must equal or exceed currently available IPV in attributes including safety, stability, antigenic integrity, immunogenicity, and affordability. With this in mind, we examined the relative attributes of several expression systems for production of PV rsVLPs (Table 1).

For each expression system the structural precursor protein, P1, of each serotype was co-expressed with the PV protease precursor, 3CD, which has a narrower substrate specificity than the mature protease, 3C, and therefore is less cytotoxic[21]. A mutation at the cleavage site within 3CD that minimised processing into 3C and 3D[44] was used in some cases (3CD*). The P1 and 3CD open reading frames were expressed separately using different strategies to suit the diverse systems as depicted in Supplementary Figs. 1a–4a and described below.

a. Mammalian expression using modified vaccinia Ankara (MVA) vectors

During PV infection, particle assembly occurs within the mammalian host cell in the presence of all viral proteins involved in replication. It was, therefore, important to determine whether recombinant particles expressed in a comparable cellular environment, but in the absence of all PV proteins except for the structural and protease precursors, had similar properties to ECs produced by natural infection. Whilst unlikely to be the system of choice for a post-eradication PV vaccine because the expression is vectored by MVA, VLPs produced in a mammalian context provide a gold standard comparison against those from other recombinant expression systems and are relevant to alternative vectoring and mRNA vaccine strategies. We previously reported the successful production of wt PV VLPs of all three serotypes and of stabilised PV3-SC8 VLPs using MVA for simultaneous expression of P1 and 3CD*[39]. We have now extended this to produce PV1 and PV2 rsVLPs (Supplementary Fig. 1a). Both the P1 and 3CD* sequences were optimised for mammalian expression. Cell contents were released by lysis into the culture supernatant for rsVLP purification (Supplementary Fig. 1b).

b. Insect cell expression

VLP vaccines against human papillomavirus (HPV) infection are produced in insect cells using the baculovirus expression vector system, providing a strong precedent for the application of this technology for human vaccines[33,35]. Recombinant baculoviruses were generated using the Bac-to-Bac system and the pFastBacDual transfer vector with the PV P1 proteins expressed from the (polyhedrin) PH promoter and the 3CD proteins from the p10 promoter (Supplementary Fig. 2a). The 3CD sequences were codon optimised for Sf9 cells

but the P1 codons were mostly wild type. Since trial expression using a P1 of PV2-SC6b with native mammalian codon usage resulted in a low level of expression this was repeated with Sf9 codon optimisation resulting in a doubling of the yield (see Table 2). Expression levels for the other two serotypes were reasonable so their P1 sequences were not codon optimised. Purification of rsVLPs was performed separately from both supernatants and lysed cells after various expression times (Supplementary Fig. 2b).

c. Yeast expression

The hepatitis B vaccine, comprising surface antigen (HBsAg) particles, has been used to immunise many millions of people since its first introduction in the 1980s and more recently an HPV VLP vaccine has also been successfully developed in this system[35,45,46]. Therefore, there is strong evidence for the effective and affordable production of licenced vaccines in yeast. In agreement with previous studies on the production of wt PV1 and other picornavirus VLPs in yeast, we found that a dual promotor expression cassette was the most efficient method of producing PV VLPs[36,47–49]. The expression cassettes were integrated into *Pichia pastoris* genomes using the *Pichia*-pink system. PV P1 with the native codon sequence and 3CD* with a *P. pastoris* optimised codon sequence were expressed from independent alcohol oxidase promoters (Supplementary Fig. 3a). After induction by the addition of methanol to the culture media, rsVLPs were purified from cell lysates (Supplementary Fig. 3b).

d. Plant expression

There is increasing interest in the use of plants as expression systems to produce pharmacological products, including vaccines. In addition, there is evidence that VLPs can be expressed efficiently in plant cells[50,51], and we have previously reported the successful production of PV3-SC8 rsVLPs in *Nicotiana benthamiana*[38]. We here expand the study to serotypes 1 and 2. Transient expression of PV rsVLP sequences was accomplished via the Agrobacterium transduction system. All PV sequences were codon optimised for expression in plants and P1 and 3CD were expressed from independent Agrobacterium clones (Supplementary Fig. 4a). Following co-infiltration into *Nicotiana benthamiana* leaves, rsVLPs were usually purified from ground leaf material after 6 days (Supplementary Fig. 4b).

For expression of the PV genes, the plant vector pEAQ-HT[38] and the mammalian vector pMVA[39] used a similar strategy for enhanced expression of P1: a viral 5'UTR and 3'UTR with the latter enhancing expression from the former. In both cases the 5'UTRs were engineered to remove an in-frame start codon for a protein upstream of the main initiation site for the viral capsid polyprotein of respectively cowpea mosaic virus (CPMV) in plant cells[52] and Foot-and-mouth disease virus (FMDV) in mammalian cells[53]. In plants, expression vector pEAQ-HT was used for expression of the P1 sequences. Reversing the 5'UTR mutation results in vector pEAQ which expresses 3CD to about 10% of the level obtained when using vector pEAQ-HT[54]. In pMVA a picornavirus IRES sequence, which is inefficiently used in BHK-21 cells, was used to modulate 3CD* expression[39].

### Particle purification

Following expression, rsVLPs were released by lysis methods appropriate for each system. After clarification to remove cellular debris, the particles were purified by a combination of chemical precipitation and/or differential ultracentrifugation and final density gradient sedimentation as outlined in Table 1 and Supplementary Figs. 1b–4b.

### Antigenic characterisation of PV rsVLPs produced in different expression systems

Following the successful production of PV rsVLPs from each expression system, we characterised the antigenic and thermostability properties of the particles (Table 2). Interestingly, although the P1 protein sequences were identical for each rsVLP serotype, there were differences in D Ag production levels depending on the recombinant

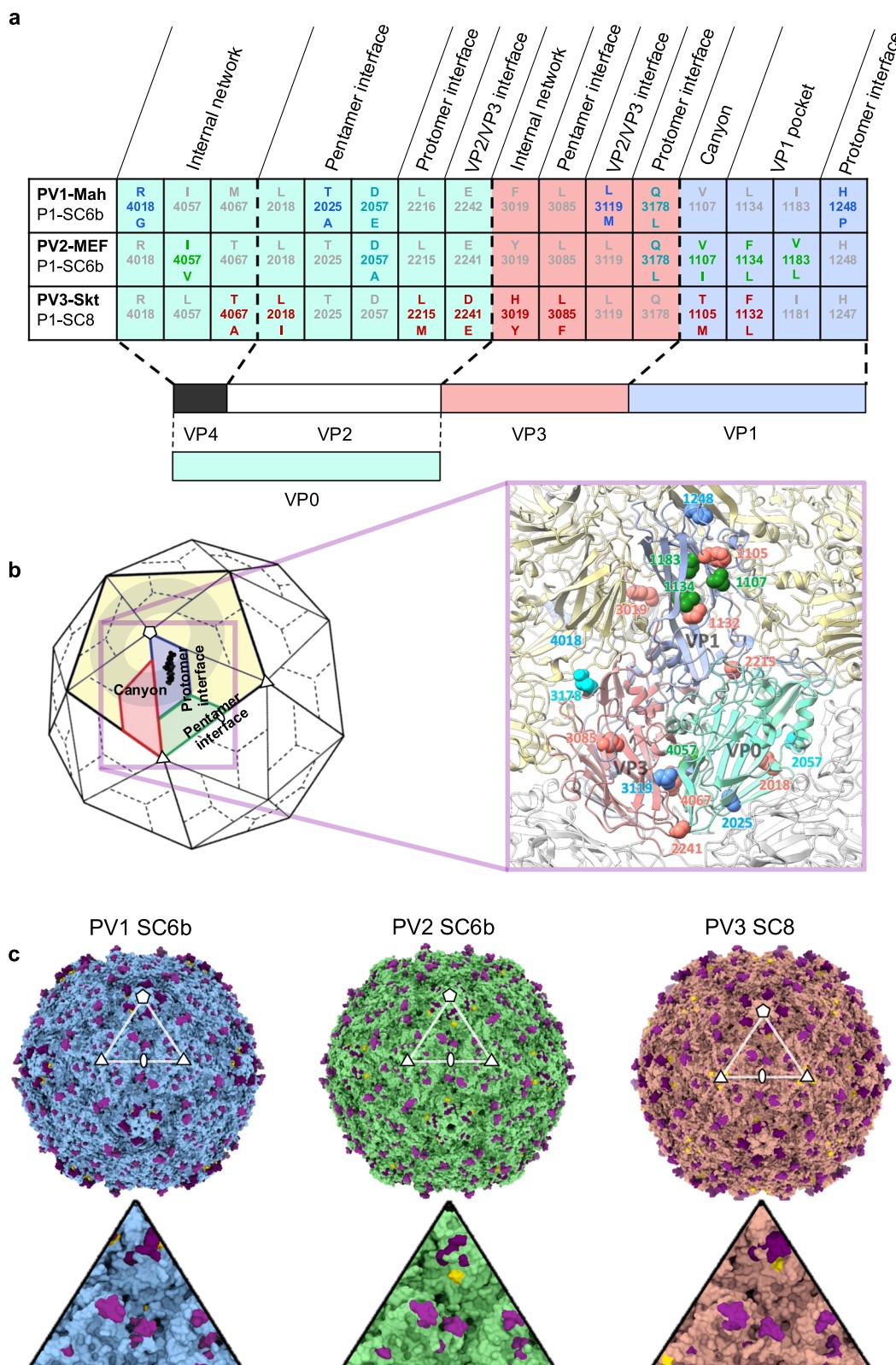

system used. Despite PV1-SC6b generating low levels of D Ag from the mammalian expression system, this mutant was produced to good levels in plants and high levels of D Ag were obtained in yeast and insect cells. PV3-SC8 was the most amenable to consistent D Ag production across all the expression systems. Conversely, PV2-SC6b produced the lowest amount of D Ag in all four cell types; PV2 requires the least D Ag per vaccine dose (32:8:28 D Ag units per dose for PV1, PV2

and PV3, respectively) and therefore the expression levels observed were suitable for further evaluation. Plant expression of PV2-SC6b yielded very low amounts of VLPs that were insufficient for evaluation of their immunological properties. Overall, in our lab-scale expression models, the yeast and baculovirus-based expression systems produced the most vaccine doses per 100 mL culture across all three serotypes (Table 2).

**Fig. 2 | Sequence and structural arrangement of stabilising mutations designed in PV rsVLPs. a** Schematic display of stabilising mutations designed for each PV serotype with a description of their location relative to capsid features. Those mutations present in each serotype are denoted in bold. The four-digit sequence numbering denotes the mature capsid subunit in the first digit (e.g. R4018G refers to VP4 R18G). **b** Cartoon of an icosahedral PV capsid, with a single pentamer shaded in pale yellow. Five-fold, three-fold and two-fold symmetry axes are labelled with symbols. A single protomeric subunit within the pentamer is highlighted and subunits coloured blue (VP1), green (VP0) and red (VP3). The lipid bound in the VP1 hydrophobic pocket is depicted in black. Key capsid features from (**a**) are labelled and the canyon around the five-fold axis is shown as a semi-transparent grey ring.

The expanded view shows the capsid protomer as a molecular cartoon with the positions of all stabilising mutations from (**a**) mapped onto the structure and colour-coded based on their insertion into the PV1-SC6b (blue), PV2-SC6b (green) and PV3-SC8 (red) rsVLPs. Mutations present in more than one rsVLP are coloured cyan. **c** The structures of serotypes PV1, PV2 and PV3 capsids (based on PDB codes 1HXS, 1EAH and 1PVC, respectively) are shown as surface representations shaded blue, green and red respectively with their antigenic sites coloured purple. Mutations introduced into the rsVLP are coloured yellow. Five-fold, three-fold and two-fold symmetry axes are labelled with symbols and an icosahedral asymmetric unit (AU) is highlighted with a white triangle. Enlarged views of each AU are presented beneath the corresponding capsids.

In addition to determining the yields of D Ag, we also assessed the quality of the antigens produced by comparing the relative levels of D:C Ag (Table 2). The impact of C Ag on the immunity induced by D Ag has not been described, however, a favourable D:C ratio would maximise the utility of the expressed proteins and reduce costs. Unfortunately, there is no PV2 C Ag-specific Mab, therefore we were only able to qualitatively assess the D:C ratio for PV1 and PV3 rsVLPs. Each expression system displayed acceptable D:C Ag ratios; in particular, baculovirus-derived rsVLPs contained no detectable C Ag, suggesting that all VLPs produced in insect cells would induce a protective immune response.

## Thermostability of PV rsVLPs produced in different expression systems

A key factor for any post-eradication vaccine will be the ability to withstand breaks in cold chains whilst maintaining the immunogenic D Ag conformation. Therefore, we assessed the thermal stability of the rsVLPs produced in all four expression systems (Table 2). Remarkably, rsVLPs displayed differing levels of thermostability depending on the expression system used. Importantly, each of the rsVLPs was stable above 40 °C with significant improvements in stability above wt ECs produced in PV-infected mammalian cells[31], ranging from a 7.5 °C improvement for PV1-SC6b expressed in yeast to a 29 °C gain in thermostability for PV3-SC8 rsVLPs made in MVA-infected mammalian cells. Furthermore, the thermostability of these rsVLPs compared favourably with the current IPV, with PV3-SC8 substantially, and PV2-SC6b modestly outperforming IPV across all expression systems; only PV1-SC6b rsVLPs from yeast, plants and insect cells were less thermally stable than IPV.

## CryoEM structure analyses of PV rsVLPs

Purified samples of PV rsVLPs produced from different expression systems were further analysed using single-particle cryoEM (Tables S1, S2). CryoEM structures of the PV1-SC6b (yeast), PV1-SC6b (MVA), PV2-SC6b (MVA) and PV2-SC6b (baculovirus) rsVLPs were determined at 3.3 Å, 3.0 Å, 2.3 Å and 2.6 Å resolution respectively (Supplementary Fig. 5). The rsVLPs were found in the D Ag conformation in all serotypes and expression systems (Fig. 3), except for PV1-SC6b produced via MVA in mammalian cells (Fig. 3a, Table 2). Only a C Ag particle could be reconstructed from PV1-SC6b expressed via MVA and few D Ag particles were observed in the data, (Fig. 3a). This was despite some D Ag potency being detected in initial preparations of MVA-produced PV1-SC6b (Table 2). Note that the conversion of D Ag particles to the C Ag form can be triggered by a number of factors and so we cannot rule out that sample preparation has increased the proportion of C Ag form observed by cryoEM. In contrast, the same rsVLPs from yeast assembled significant numbers of D particles (1320 picked particles were D and 2428 were C antigenic form, a ratio of 1:1.8). The C Ag particles for PV1-SC6b (yeast and MVA) exhibited distinctive holes at their 2-fold symmetry axes compared with the D Ag particle of the same rsVLP from yeast (Fig. 3a), in line with observations of the expanded state induced in PV virions[28]. In the case of PV2-SC6b from MVA-mediated

expression in mammalian cells and baculovirus-mediated expression in insect cells, although there is no C Ag specific ELISA, cryoEM analysis indicated the absence of C Ag particles. The PV1-SC6b and PV2-SC6b D Ag structures of rsVLPs were essentially identical in overall conformation to their mature D Ag PV1 and PV2 virion counterparts, with an RMSD in Cα of 0.70 Å and 0.45 Å, respectively. Furthermore, the stabilising mutations did not introduce any detectable local changes in the antigenic surface residues; the average RMSD in Cα atoms between D Ag PV1-SC6b (yeast) and PV1 virion (PDB 1HXS) was 0.47 Å over antigenic sites, and that between D Ag PV2-SC6b from MVA and baculovirus against PV2 virion (PDB 1EAH) was 0.51 Å and 0.49 Å, respectively. However, as expected for VLPs with unprocessed VP0[23] some of the internal regions were less ordered compared to mature virions; on average ~57 residues were disordered at the N-terminus of VP0 across the PV1-SC6b and PV2-SC6b structures reported here, which corresponds to the majority of the VP4 peptide in mature PV virions. Surprisingly, however, we observed an unexpected chain break between residues 43 and 45 of VP0 for the PV2-SC6b structure produced from mammalian cells, whereas cleavage occurs after residue 69 in virions. It appears that this cleavage allowed residues 45 to 49 to adopt an alternative conformation to that seen in the mature capsid (PDB 1EAH), placing residue 45 ~25 Å from residue 43.

There were no significant structural differences between the D Ag rsVLPs across serotypes expressed in different systems, with an average RMSD in Cα of 0.70 Å between the D Ag PV1-SC6b rsVLP from yeast and PV2-SC6b rsVLPs from MVA and baculovirus. However, since the stability of the D Ag particles is enhanced by lipidic moieties within the internal VP1 pocket[55] we carefully evaluated the density seen in this pocket in the rsVLPs (Fig. 3b). The D Ag structures for the mammalian cell (PV2-SC6b), insect cell (PV2-SC6b) and yeast cell (PV1-SC6b) expressed particles all had essentially fully occupied lipid pockets and the unambiguous cryoEM potential maps suggested that there was little chemical difference in the molecules (or mix of molecules) occupying them (Fig. 3b–d). However, whilst the length of observed cryoEM density in the PV2-SC6b structures (MVA and baculovirus produced) indicated that a lipid such as sphingosine was present (18 carbon length, Fig. 3c, d) the equivalent VP1 pocket density for the D Ag PV1-SC6b from yeast indicated a shorter lipid moiety, modelled here as palmitic acid (16 carbon length) (Fig. 3b left panel). However, the lower resolution of the D Ag PV1-SC6b particle from yeast (3.3 Å) compared to the D Ag PV2-SC6b particles from mammalian and insect cells (2.3 Å and 2.6 Å, respectively) means that this should be interpreted with caution. In the case of the expanded C Ag structures observed for PV1-SC6b produced in yeast and mammalian cells, the VP1 pocket was in a collapsed state compared to that of D Ag particles (Fig. 3b); filled by sidechain residues Ile-157, Tyr-159 and Phe-237 of VP1 and consequently empty of any observable cryoEM density indicating bound lipid (Fig. 3b right panel). As previously reported for PV3-SC8 rsVLPs from plants[38,56] the lipid was readily replaced in the D Ag PV1-SC6b particle from yeast by incubating a molar excess of the tailored high affinity pocket binder GPP3 (VLP:compound molar ratio of 1:300) (Fig. 3e). It was also possible to simultaneously bind glutathione (GSH)

**Table 1 | Summary of expression systems**

| Expression system | Transfer plasmid /vector | Cells | Incubation temperature | Culture volume /Harvest time | Initial processing | Clarification at medium speed | Intermediate concentration | Sucrose cushion percentage /centrifugation parameters | Gradient purification |
|---|---|---|---|---|---|---|---|---|---|
| Mammalian cells | p434 /MVA | BHK-21 | 30 °C | 2–5 x T175 /12 h | Freeze thawing | 10,000 × g 20 min, 4 °C | | 2 ml of 30% 145,600 × g for 5 h, 4 °C | 15–45% sucrose 75,300 × g 22 h, 4 °C |
| Insect cells | pFastBacDual /baculovirus | Sf9 | 27.5 °C | 150–200 ml /3–7 dpi | 0.5% NP40 | 10,000 × g 20 min, 4 °C | Optional 8% PEG 8000 concentration of supernatant | 2 ml of 30% 145,600 × g for 5 h, 4 °C | 15–45% sucrose 75,300 × g 22 h, 4 °C |
| Yeast cells | pPink Dual /n.a. | Pichia pastoris | 28 °C | 2 ml for 48 h → 200 ml for 24 hrs + 48 h after methanol induction | 0.1% Triton-X 100 Cell disruptor | 10,000 × g 30 min, 4 °C | 8% PEG 8000 precipitation of lysate | 3 ml of 30% 151,000 × g for 3.5 h, 10 °C | 15–45% sucrose 151,000 × g 3 h, 10 °C |
| Plant cells | pEAQ-HT and pEAQ /Agrobacterium Tumefaciens LBA4404 | Nicotiana benthamiana | 25 °C | 10–100 g of leaf material/6 dpi | Grinding of leaves in homogeniser | Coarse and fine filtration of grindate 9500 × g 15 min, 4 °C | Ultrafiltration | 1 ml 70% and 5 ml 25% 167,000 × g for 3 h, 4 °C | 20–60% Nycodenz 247,103 × g for 24 h, 4 °C |

at 10 mM into the interprotomer surface pocket of the same rsVLP, to form the ternary complex of yeast-derived PV1-SC6b[GPP3+GSH]. This complex yielded a 2.8 Å reconstruction as assessed by the FSC 0.143 threshold (Fig. 3e and Supplementary Fig. 5a). The cryoEM electron potential map revealed the PV1-SC6b[GPP3+GSH] rsVLP from yeast to be well-ordered in the D Ag conformation (Fig. 3e) and there was an unambiguous feature for bound GSH in the pocket formed between VP1 subunits from two adjacent protomers and VP3 from a single protomer (Fig. 3e, f). In addition, the VP1 hydrophobic pocket was occupied by GPP3 in a conformation consistent with that previously observed for this compound bound to PV3-SC8 from both plant[38] and yeast[56] cells (Fig. 3g). We have previously shown binding of GSH at the interprotomer interface for the PV3-SC8 rsVLP from yeast[56]. The structure of yeast-derived PV1-SC6b[GPP3+GSH] reported here demonstrates that the mode of binding of GSH is essentially identical for both serotypes (Fig. 3f), with an RMSD in Cα between the PV1-SC6b[GPP3+GSH] complex and the PV3-SC8[GPP3+GSH56] complex of 0.53 Å. Interestingly, upon incubation with stabilising additives such as GSH and GPP3, the PV1-SC6b rsVLPs from yeast were observed to be entirely D Ag by cryoEM analysis, with no significant population of C Ag particles (Fig. 3e), which further supports our previous report of the stabilising effect of rsVLPs by GPP3 and GSH[38,56].

## Immunogenicity

Following antigenic and structural characterisation of the rsVLPs, we assessed their immunogenicity through a PV challenge model using transgenic mice expressing the PV receptor (referred to as TgPVR). Following intraperitoneal immunisation with 2 × 0.5 human doses of D Ag for PV1-SC6b, PV2-SC6b and PV3-SC8, (16:4:14 D Ag units respectively) or 2 × 0.5 human doses of IPV, mice were bled to assess the level of seroconversion induced by the rsVLPs/IPV. Irrespective of the expression system used, all the rsVLPs elicited similar or better neutralising antibody responses compared to the IPV reference (Fig. 4).

The immunised TgPVR mice were challenged with 25× PD$_{50}$ of virus of the corresponding serotype following either a single or two injection(s) with 0.5 human doses of rsVLPs or IPV and monitored for 14 days. Sera were collected pre-boost and pre-challenge and the virus neutralising antibody levels compared to those of mice inoculated with a PBS control or the IPV standard (Fig. 5a). Using yeast-derived PV1-SC6b as an exemplar, a single immunisation with rsVLPs induced a higher neutralising antibody response and protected all mice in the group against virus challenge, compared with the group immunised with the equivalent dose of IPV, which had a lower detectable neutralising antibody titre and protected only 3/8 mice against virus challenge. Following 2 injections with IPV, the neutralising antibody titre increased and all of the group were protected against challenge. 7 mice boosted with a second dose of rsVLPs also showed an increase in neutralising antibody titre, and were again, all protected against virus challenge (Fig. 5b).

In addition to the TgPVR mice challenge model, any future PV vaccine will need to pass the pharmacopeial IPV lot release assay, developed at MHRA using Wistar rats[57]. In this model, rats were immunised IM with either IPV or rsVLPs at doses ranging from 1 to 0.125 human dose and the resulting sera assessed for neutralising antibody titres (Fig. 6a). In rats, PV3 rsVLPs produced in plants and yeast were as immunogenic as IPV whereas those produced in insect and mammalian cells were notably more immunogenic (Fig. 6a). The immunogenicity of PV1 and PV2 rsVLPs in rats was inferior to that of IPV (Fig. 6a), irrespective of the recombinant expression system used for production, in contrast to the findings in the TgPVR mice challenge study.

Since all currently licensed VLP vaccines are administered alongside an adjuvant to boost the antigen responses[58], we assessed the effect of an aluminium hydroxide adjuvant on immune responses elicited in rats by rsVLPs from the yeast and baculovirus expression

**Table 2 | VLP yield and thermostability**

| Expression System | Serotype | VLP Yield (D Ag/100 mL Culture) | Vaccine Doses (per 100 mL Culture) | Thermostability (Temp °C 50% D Ag loss) | Thermostability vs WT Empty Capsids (°C) | Thermostability vs IPV (°C) | D:C Ratio |
|---|---|---|---|---|---|---|---|
| Mammalian | PV1-SC6b | 26.3 | 0.8 | 49 | +13 | 0 | D>C |
| | PV2-SC6b | 76.5 | 9 | 57 | +15 | +5 | - |
| | PV3-SC8 | 698 | 25 | 62 | +29 | +10 | D>>C |
| Baculovirus | PV1-SC6b | 398 to 969 | 12–30 | 44 | +8 | −5 | D only |
| | PV2-SC6b | 127 to 138[a] | 16–18 | 49.5 | +7.5 | 2.5 | - |
| | PV3-SC8 | 935 to 2267 | 33–81 | 60 | +27 | + 8 | D only |
| Yeast | PV1-SC6b | 1030 (±40) | 32 | 43.5 | +7.5 | −5.5 | D>>C |
| | PV2-SC6b | 69 (±12) | 8 | 52.5 | +10.5 | +0.5 | - |
| | PV3-SC8 | 745 (±65) | 26 | 55 | +22 | +3 | D>C |
| Plant* | PV1-SC6b | 235 (±71) | 7 | 44 | +8 | −5 | D>C |
| | PV2-SC6b | 1.3 (±0.4) | 0.16 | 53 | +11 | +1 | - |
| | PV3-SC8 | 180 (±28) | 6 | 54 | +21 | +2 | D>C |

*D Ag and Vaccine doses per 10 g leaf material.
[a]Before codon optimisation 63, after Sf9 codon optimisation 138 DAg/100 ml.

systems (Fig. 6b). With adjuvant, all rsVLPs tested induced significantly higher antibody responses, with PV1 and PV2 rsVLPs displaying similar or improved immunogenicity compared to IPV whereas PV3 rsVLPs induced significantly superior responses to IPV.

Therefore, in the presence of an adjuvant used in currently licenced pentavalent and hexavalent childhood vaccines, rsVLPs of all three serotypes were at least as immunogenic, per DAgU, as IPV.

## Discussion

Due to the success of vaccination campaigns, wt PV2 and wt PV3 have been eradicated with only wt PV1 now circulating in a few countries[2,3]. Unfortunately, due to the use of OPV, cVDPV/iVDPV cases now outnumber those from wt PV[2]. Complete elimination of PV therefore requires alternative vaccines obviating the need to culture infectious virus. Here, our WHO-funded consortium has shown that PV rsVLPs that maintain the D Ag conformation can be successfully produced in four different expression systems and their potential as vaccines was assessed in animal models.

This work builds on our previous research identifying and combining stabilising mutations for each PV serotype[31]. Based on this, we selected a single set of stabilising capsid mutations for each serotype, PV1-SC6b, PV2-SC6b and PV3-SC8, respectively (Fig. 2). The construct formats were similar for all three up-scalable expression systems (insect, yeast and plant) with separate promoters used for the expression of the P1 and 3CD ORFs, whilst for MVA-mediated expression in mammalian cells P1 was under the control of the T7 promoter and 3CD was translated from an IRES. These strategies ensured that in all four cases, authentic mature capsid proteins were expressed with no extraneous sequences added to the N- or C-termini. Together with stabilising mutations that do not alter antigenicity this guaranteed that the VLPs produced reflect the ECs generated during infection in structure but not in instability. Purification of the rsVLPs will remove 3CD which is therefore unlikely to present a toxicity problem, however, PV rsVLPs have been produced in insect cells in the absence of 3CD via expression of separate VP1 and VP3-2A-VP0 ORFs, resulting in VP3 proteins to which a 2A sequence derived from porcine teschovirus remained linked, whilst VP0 included an additional N-terminal proline residue[37]. In yeast, using this strategy for expression of the three structural proteins in all possible combinations showed that only VP3 can tolerate extension with the 2A peptide and still assemble D Ag particles (with reduced yields in comparison to constructs including 3CD)[59].

The PV3 rsVLPs were the most consistent in terms of D Ag yield across all expression systems, whereas PV2 rsVLPs were the least productive. Interestingly there were some differences in both antigen yield and thermostability between expression systems (Table 2), which may be explained through features of translation such as codon usage[60–62] and the influence of enhancer sequences on the rate of translation[63], impacting the efficiency of folding, processing, and assembly of viral particles. Moreover, rsVLPs from different expression systems displayed different D:C Ag ratios and further work will be required to optimise these expression systems for industry-scale production.

PV vaccines must be thermally stable to address storage problems and the challenges of maintaining a cold-chain for delivery to remote places within low- and middle-income countries. All rsVLPs had significantly improved thermostability over wt ECs. Additionally, the thermostability of rsVLPs compared favourably to IPV, aside from PV1 rsVLPs. The fact that rsVLPs of the same serotype display different levels of thermal stability depending on the expression system used (Table 2) may be partially explained by the nature, presence or absence of pocket factor, a host-derived lipid which binds into the VP1 β-barrel in infectious particles and is important for capsid stability[55]. Our cryoEM structures showed that a pocket factor is present in rsVLPs from mammalian, insect and yeast cells[42] but absent from plant-expressed rsVLPs[38] but otherwise the structures of the D Ag particles were essentially identical for all expression systems (Fig. 3).

GSH, which binds within a VP1-VP3 interprotomer pocket further increases thermostability[56]. The 2.8 Å cryoEM structure of the yeast-derived PV1-SC6b[GPP3+GSH] complex reported here confirms that the GSH binding mode is conserved across serotypes when compared with the PV3-SC8[GPP3+GSH] complex reported earlier[56]. The addition of synthetic pocket factor compounds has been shown to promote native antigenicity in VLPs produced in *Saccharomyces cerevisiae*[41]. The structure of the yeast-derived PV1-SC6b[GPP3+GSH] complex containing bound GPP3 in the VP1 hydrophobic pocket and GSH at the VP1-VP3 interprotomer pocket displayed an entirely D Ag conformation in contrast to the mixed D/C Ag population of the unbound form of the same rsVLP (Fig. 3a). Since GSH and GPP3 bind synergistically at distinct sites on the PV rsVLPs, the development of possible stabilising excipients warrants further study.

Next-generation PV vaccines will need to elicit the same long-lasting immunity against disease as the current vaccines. Therefore, we assessed the immunogenicity of the rsVLPs in two animal models, a TgPVR mouse challenge model and the Wistar rat pharmacopeial IPV lot release assay. In the TgPVR mouse model responses to rsVLPs were superior to those obtained using IPV (Figs. 4, 5a) and this correlated with increased protection against virus challenge following

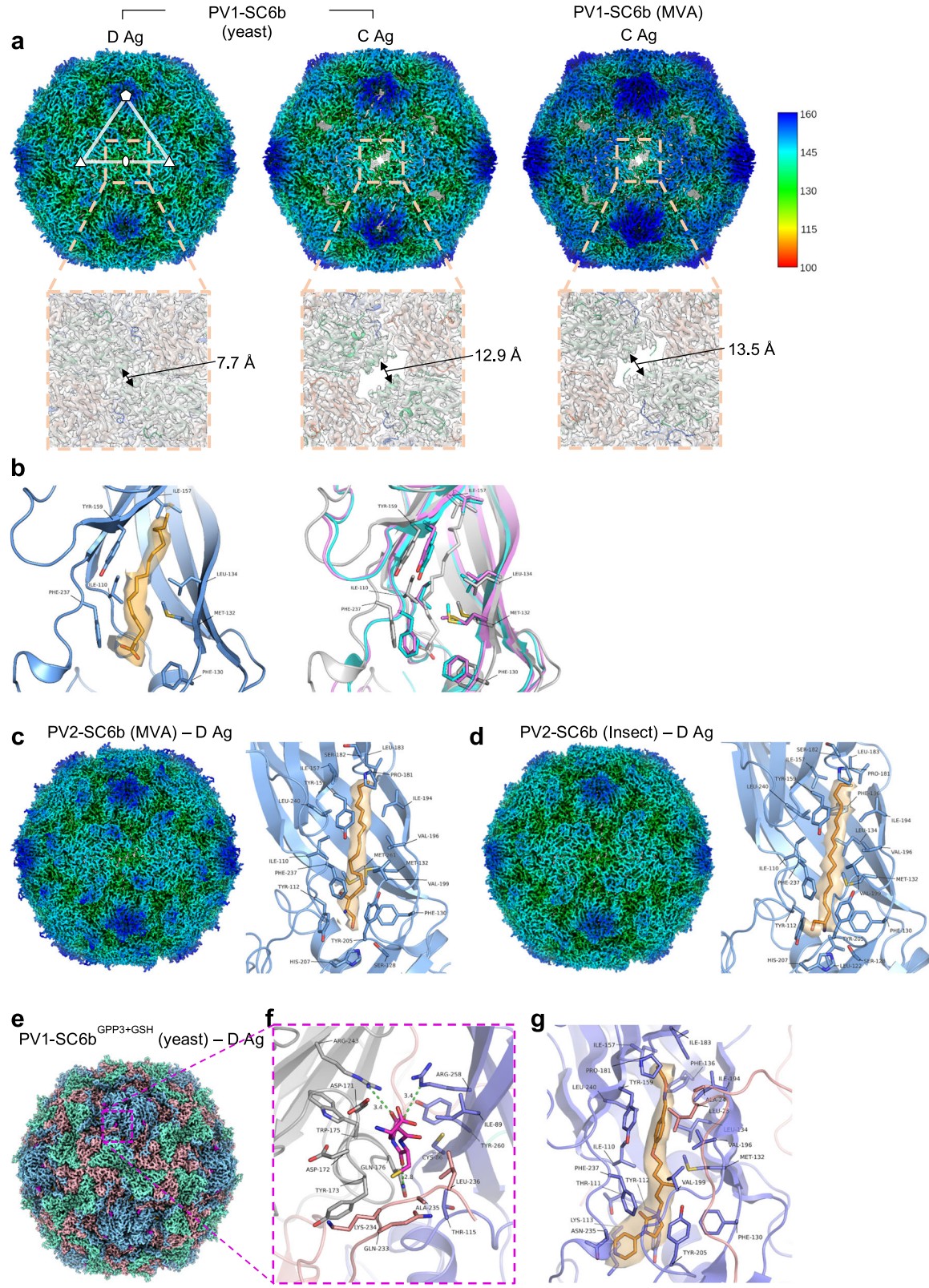

immunisation with a single dose compared with IPV (Fig. 5a, b). This may be due to the (i) increased thermostability of the rsVLPs over IPV in the case of PV2 and PV3, although PV1 rsVLPs were less thermostable than IPV (Table 1), (ii) the absence of formalin treatment, a process known to modify antigenic structure[16], possibly resulting in a broader neutralising antibody response and improved protection in the TgPVR mice.

In contrast, in the rat pharmacopeial IPV lot release assay only PV3 rsVLPs resulted in a superior neutralising antibody response to IPV whereas PV1 and PV2 rsVLPs were inferior to IPV in the absence of adjuvant (Fig. 6a). We considered whether the different routes of immunisation (intraperitoneal for mice and intramuscular for rats) may have influenced the neutralising antibody response in the respective models. However, upon comparison in the TgPVR model we

**Fig. 3 | CryoEM reconstructions of PV1-SC6b and PV2-SC6b rsVLPs and the PV1-SC6b[GPP3+GSH] complex. a** Top panel, PV1-SC6b rsVLPs from yeast and mammalian (MVA) cells depicted as isosurface representations of the electron potential maps at a threshold of 4σ (σ is the standard deviation of the map) and radially coloured by distance (Å) from the particle centre according to the colour key. Representative five-fold, three-fold and two-fold symmetry axes, and an icosahedral AU are shown as in Fig. 2c. Bottom panel, zoomed-in views of the two-fold interface of each rsVLP. **b** Left panel, cartoon depiction of the VP1 hydrophobic pocket of the yeast PV1-SC6b D Ag particle, with palmitic acid shown as an orange stick model fitted into the cryoEM map (1.0 σ). Amino acid residues interacting with palmitic acid are labelled. Right panel, VP1 pockets of the C Ag particles from yeast (cyan) and mammalian cells (magenta) are superposed on the D Ag particle VP1 pocket (grey). CryoEM maps at 4σ of PV2-SC6b rsVLP expressed in mammalian (**c**) and insect (**d**) cells, respectively (radial colouring same as **a**), shown alongside their respective VP1 pockets, with sphingosine (SPH) modelled as orange sticks. Electron potential for SPH is 1.2 σ in PV2-SC6b MVA (**c**) and 1.0 σ in PV2-SC6b baculovirus (**d**). **e** CryoEM reconstruction of the PV1-SC6b[GPP3+GSH] complex viewed along the icosahedral two-fold axis with VP1, VP0 and VP3 subunits coloured as in Fig. 2b, and GSH in magenta. **f** Zoomed-in view of GSH (magenta stick model) bound in the VP1-VP3 inter-protomer surface pocket between neighbouring capsid protomers (A and B) of PV1-SC6b[GPP3+GSH]. VP1 and VP3 of protomer 'A' are coloured light blue and light red, respectively. VP1 of protomer 'B' is coloured grey. Residues of VP1 and VP3 forming the GSH binding pocket are shown as sticks and labelled. Hydrogen bond and salt-bridge interactions are shown as green dashed lines and distances labelled. **g** PV1-SC6b[GPP3+GSH] rsVLP VP1 pocket with GPP3 fitted into the cryoEM map (1.5σ) as an orange stick model and interacting residues labelled. All cryoEM maps are rendered at a radius of 2 Å around depicted atoms.

observed no significant difference in neutralising antibody titres induced following immunisation with IPV or rsVLPs via intraperitoneal or intramuscular administration (Supplementary Fig. S6). Therefore, the observed contrast may reflect inherent differences in the immune systems of the two animal models, as highlighted by a recent study of the polyclonal sera response to coxsackievirus B3 infection in mice and in humans, which revealed mice to have a restricted response to a single antigenic site compared to a broader more diverse response in human sera, targeting a number of sites across the capsid[64]. Another possibility is that the viral RNA, which is present in IPV, but we have previously shown to be absent in VLPs[36,42], acts as an adjuvant in rats but is less potent in mice. Therefore, since licensed VLP vaccines are administered with adjuvant[58], we inoculated rats with rsVLPs in the presence of an adjuvant used in currently approved multivalent childhood vaccines (Fig. 6b). With adjuvant rsVLPs of all three serotypes were at least as immunogenic, per human dose, as IPV and would pass the pharmacopeial lot release assay[65].

In conclusion, we have demonstrated that rsVLPs from all three PV serotypes can be produced from scalable expression systems whilst maintaining antigenic integrity and in the presence of adjuvant these rsVLPs are as, or more immunogenic than the current IPV. Although further work is required to investigate scalability of production, overall, our data shows that our PV rsVLPs are excellent candidates to produce post-eradication vaccines as, crucially, they do not require the growth of infectious PV.

## Methods
### Expression and purification of PV rsVLPs from mammalian cells using recombinant modified vaccinia Ankara (MVA)
We have previously reported the construction of the P1-3CD expression cassette for the PV3-SC8 rsVLP, and introduction into a recombinant MVA vector[39]. Briefly for PV1-SC6b and PV2-SC6b, gene sequences for the P1 region were codon optimised for mammalian cells and cloned into transfer vector pMVA upstream of a 3CD sequence taken from PV1 Mahoney comprising an additional Ser at the 3C-3D junction to restrict 3CD processing (3CD*). In the resulting dicistronic cassette, expression of 3CD* was regulated using the PV3-Leon internal ribosome entry site (PV-IRES). Additional elements of the expression cassette included the T7 promoter and an FMDV IRES upstream of P1, the FMDV 3'UTR downstream of 3CD followed by a 20-nucleotide long polyA tail and the T7 terminator element[39] (Supplementary Fig. 1).

PV rsVLPs were produced by dually infecting BHK-21 cells with an MVA-PV recombinant virus harbouring the P1-3CD* cassette and an MVA-T7 virus expressing T7 polymerase. The infection was left to proceed at 30 °C for 12 h. Cell suspensions were lysed by freeze-thawing into the culture medium, and VLPs were purified from clarified supernatants by concentration through a 30% (w/v) sucrose cushion followed by a 15–45% (w/v) sucrose gradient in 1 × Dulbecco's phosphate-buffered saline (DPBS, Gibco), 20 mM EDTA pH 7.0 (DPBS-EDTA)[39] (Table 1 and Supplementary Fig. 1).

### Expression and purification of rsVLPs from insect cells
Synthetic PV-specific gene fragments encoding 3CD (Genscript) derived from PV1 Mahoney, PV2 MEF-1 and PV3 Saukett were codon optimised for expression in Sf9 cells and cloned into plasmid pFastBac Dual (Invitrogen) downstream of the p10 promoter using restriction sites *Nco*I and *Kpn*I. Resulting p10-3CD plasmids were digested with *EcoR*I and *Hind*III for insertion after the PH promoter of the P1 gene from the serotype matching the 3CD sequence. These P1 genes encoded capsid mutants PV1-SC6b, PV2-SC6b and PV3-SC8 using unmodified codon sequences and had been amplified by PCR from plasmids pT7RbzLeonMahP1_SC6b_HindIII del, pT7RbzLeonMEF2P1_SC6b_HindIII del and pT7RbzLeonSktP1_SC8_NdeI del generated at MHRA. These three plasmids comprise large deletions that encompass most of the polymerase gene of the full-length PV genomes rendering them safe for handling in non-containment lab facilities. PCR primers for P1 amplification were designed either with *EcoR*I and *Hind*III endings for restriction cloning or with 15b overlaps for In-Fusion® HD cloning (Takara) as required. In addition, a construct expressing P1 of PV2-SC6b was made with a sequence optimised for expression in Sf9 insect cells (GeneArt) (Supplementary Fig. 2).

All four PH-P1_p10-3CD expressing pFastBac Dual constructs were used to generate corresponding baculoviruses, namely bac-PV1-SC6b, bac-PV2-SC6b, bac-PV2-SC6b (Sf9opt) and bac-PV3-SC8. Recombination into bacmid employed the Bac-to-Bac system (Invitrogen), used according to the manufacturer's instructions.

PV rsVLPs were produced by infecting Sf9 insect cells. Cells and supernatants were harvested either 3 or 7 days after infection. Centrifugation at $1000 \times g$ for 15 min at 4 °C produced cell pellets which were frozen at −20 °C. Supernatants were filtered using 0.22 µm filtering devices (Steritop, Millipore). Thawed cell pellets were lysed into DPBS + 20 mM EDTA pH 7.0 containing 0.5% Igepal (CA-630, Sigma-Aldrich) and protease inhibitors (P5884, Sigma-Aldrich) on ice for 30 min followed by clarification at $10,000 \times g$ for 20 min at 4 °C. Re-extraction of the cell debris pellets with 1 volume of chloroform produced an aqueous extract that was pooled with the clarified lysates. Filtered supernatants and clarified lysates were pelleted through a 30% sucrose cushion in DPBS + 20 mM EDTA by ultracentrifugation at $145,600 \times g$ in an SW32 rotor (Beckman) for 5 h at 4 °C. Pellets were resuspended in DPBS + 20 mM EDTA o/n, cleared by centrifugation at $10,000 \times g$ for 15 min at 4 °C in a microfuge, and loaded onto 15–45% (w/v) sucrose density gradients. Centrifugation was at $75,600 \times g$ for 22 h at 4 °C using an SW41 rotor (Beckman). Gradients were fractionated by manual collection from the top and fractions containing VLPs were identified by western blot analysis using a blend of monoclonal antibodies for the detection of VP1 (MAB8566, Sigma-Aldrich) (Table 1 and Supplementary Fig. 2).

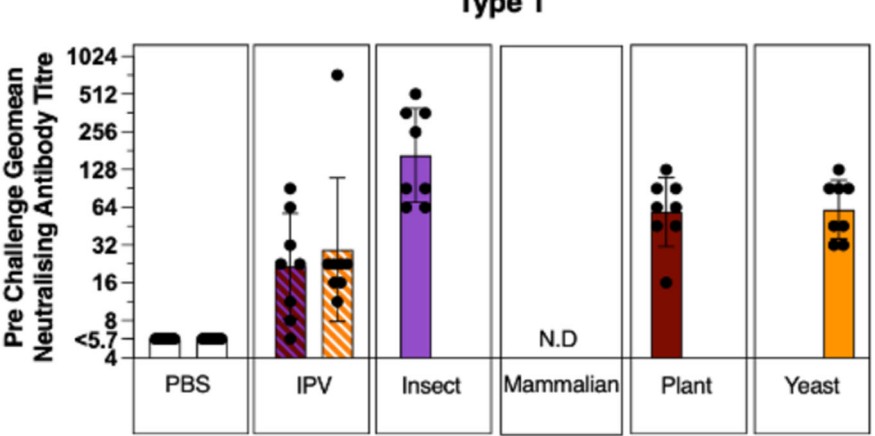

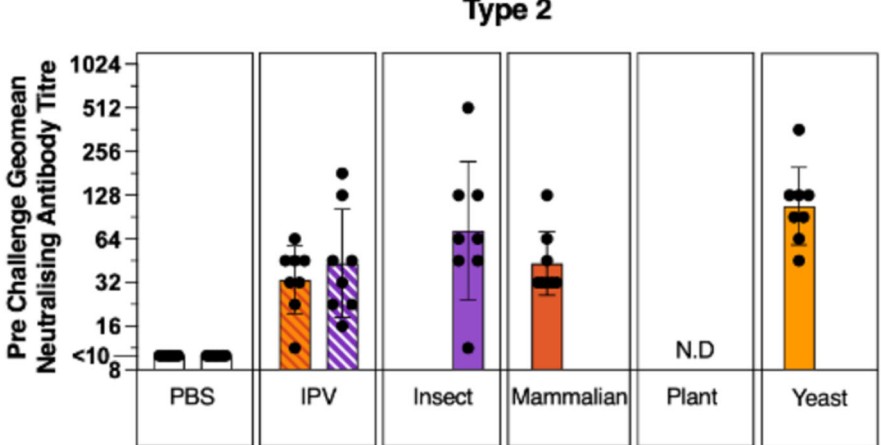

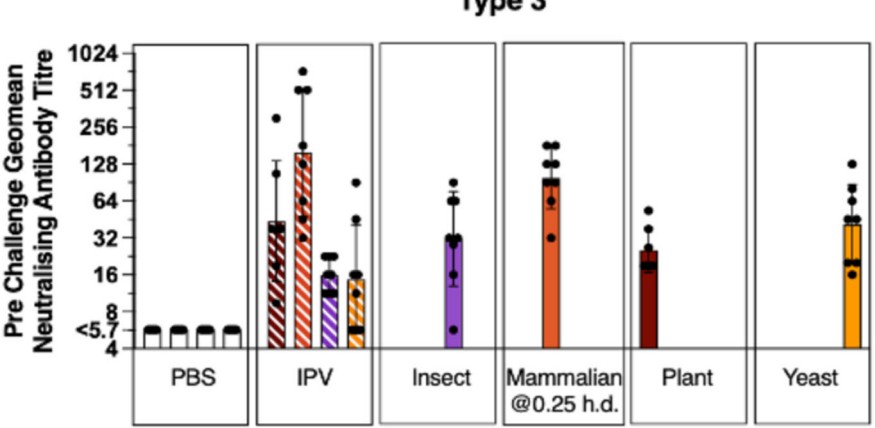

**Fig. 4 | Immunogenicity of rsVLPs in TgPVR mice.** Neutralising antibody titres following immunisation with VLPs from each of the 3 PV serotypes produced in different expression systems. Groups of 8 mice received 2 injections (on days 0 and 14) of 0.5 human doses of either PV1-SC6b (16 D Ag), PV2-SC6b (4 D Ag), PV3-SC8 (14 D Ag), IPV or PBS as the negative control. Sera were collected 35 dpi and neutralisation assays performed as described[91]. Each IPV bar is representative of a different experiment positive control. The position of each bar and the colour of the stripes in the IPV bars indicate for which expression system(s) IPV was the positive control (Insect; purple, Mammalian; red, Plant; brown, and Yeast; orange). N.D indicates the experiment was not done due to insufficient D Ag. Error bars represent the Geomean Standard Deviation of the data points. Source data are provided as a Source Data file.

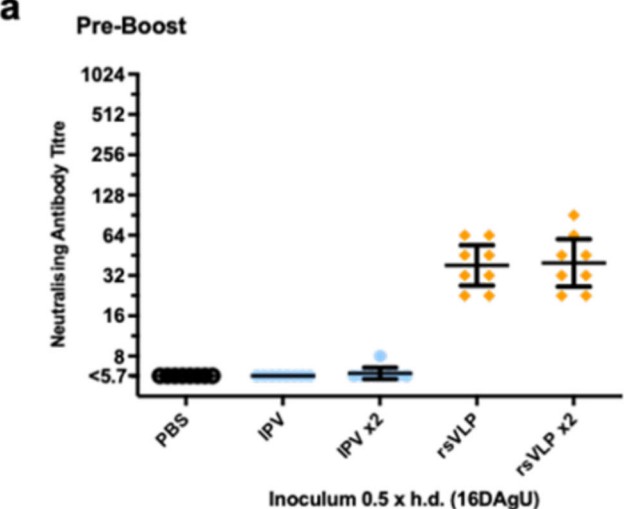
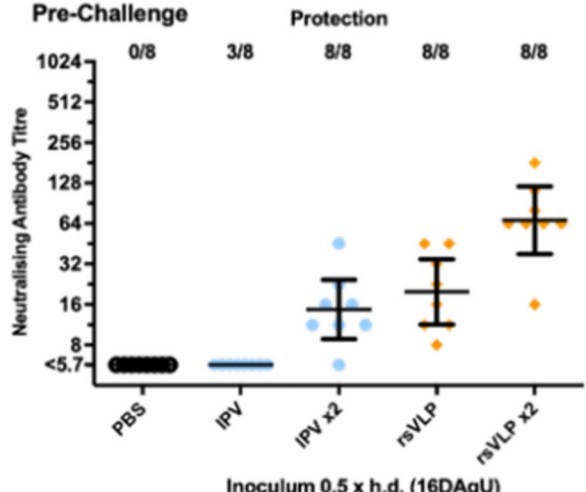

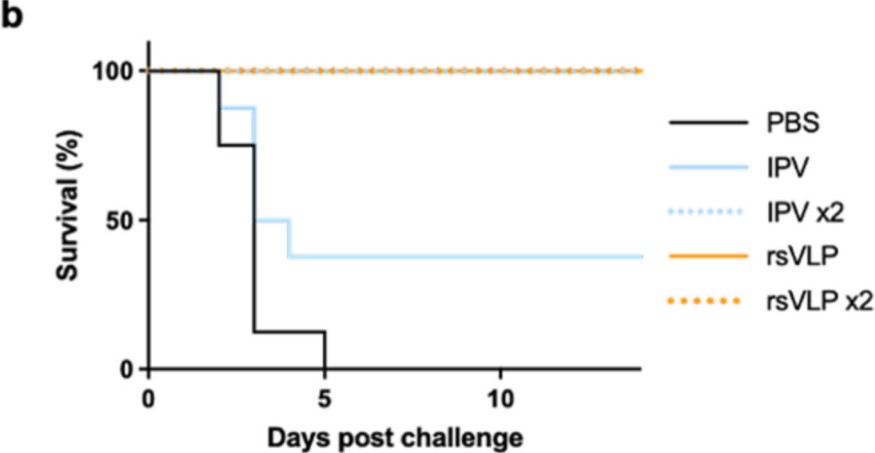

**Fig. 5 | Protection against live challenge in TgPVR mice immunised with PV1-SC6b VLPs. a** Groups of eight mice received either a single or two doses of rsVLPs (orange) produced in yeast (day 0 and 14) prior to challenge on day 35 with $25 \times PD_{50}$ dose of live PV1 Mahoney and compared to IPV (blue) or a PBS negative control (black). Neutralisation titres prior to boost and on the day of challenge were determined. **b** Animals were monitored for survival for 14 days following challenge. Error bars represent the Geomean Standard Deviation of the data points. Source data are provided as a Source Data file.

## Expression and purification of rsVLPs from yeast (*Pichia pastoris*)

The dual promoter constructs for producing PV1-SC6b and PV3-SC8 rsVLPs have been described previously[42,59]. The P1 gene of PV2-SC6b was amplified by polymerase chain reaction (PCR) from pT7RbzLeonMEF2P1_SC6b_HindIII del. The 3CD* gene was codon-optimised for *P. pastoris* and included a cleavage-preventing mutation to reduce the potential toxic effects of 3C[44]. Both genes were cloned separately into the pPink-HC expression vector multiple cloning site (MCS) using *Eco*RI and *Fse*I (New England Biolabs (NEB)). In pPink-HC, expression of foreign genes is regulated by the methanol-induced AOX1 promoter. Subsequently, a dual promoter expression vector was constructed through PCR amplification from position 1 of the 3CD pPink-HC to position 1285, adding a *Sac*II restriction site at both the 5′ and 3′ end of the product. The P1 expression plasmid was linearised by *Sac*II (NEB), and the 3CD PCR product inserted. The resulting plasmid was linearised by *Afl*II digestion (NEB) and transformed into Pichia-Pink™ Strain one (Invitrogen, USA) by electroporation. All PCR steps

were carried out with Phusion polymerase (NEB) using the manufacturer's guidelines (Supplementary Fig. 3).

Transformed yeast cells were screened for high-expression clones by small-scale culture experiments (5 ml), with levels for each clone determined by immunoblotting using a blend of monoclonal antibodies for the detection of VP1 (MAB8566, Sigma-Aldrich). For VLP production, cultures were grown to high density in 200 ml YPD in 2L baffled flasks. After 24 h, the cells were pelleted at $1500 \times g$ and resuspended in YPM (methanol 0.5% v/v) to induce protein expression and cultured for a further 48 h at 28 °C. Cultures were fed an additional 0.5% v/v methanol at 24 h post-induction. At 48 h post-induction, cells were pelleted at $2000 \times g$ and resuspended in breaking buffer (50 mM sodium phosphate, 5% glycerol, 1 mM EDTA, pH 7.4) and frozen prior to processing.

Cell suspensions were thawed and lysed using a CF-1 cell disruptor at ~275 MPa chilled to 4 °C following the addition of 0.1% Triton-X 100. The resulting lysate was clarified through multiple rounds of centrifugation and a chemical precipitation step as previously described[42].

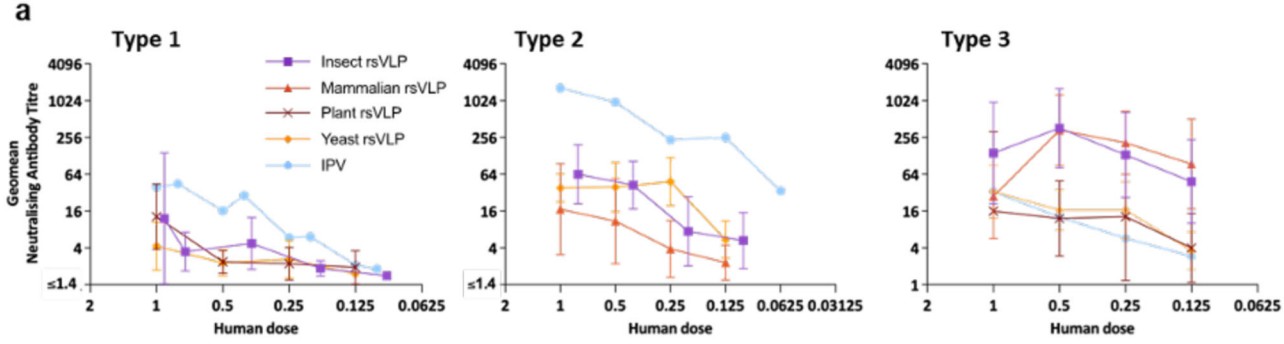

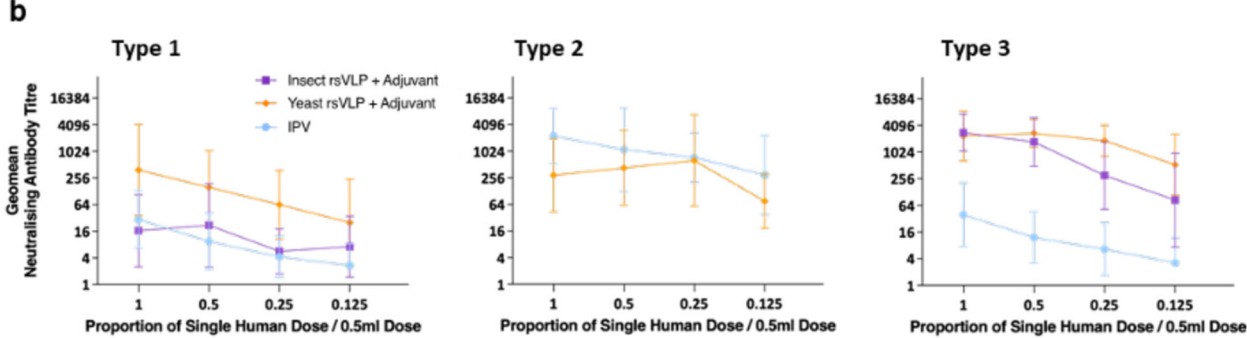

**Fig. 6 | Immunogenicity of rsVLP vaccines in female rats.** Dose response in neutralising antibodies following a single immunisation of female Wistar rats in the absence (**a**) or the presence (**b**) of adjuvant. Groups of 10 rats received rsVLPs at various multiples of human doses and compared with IPV (blue). Sera were collected 21 dpi and neutralisation titres against the Sabin strains of PV1, PV2 or PV3 were determined. VLPs produced from each expression system are depicted as follows: Insect; purple, Mammalian; red, Plant; brown, and Yeast; orange. Error bars represent the Geomean Standard Deviation of the data points. Source data are provided as a Source Data file.

The clarified supernatant was concentrated through a 30% sucrose cushion. The resulting pellet was resuspended overnight in PBS + 1% NP40 (Merck) + 0.5% sodium deoxycholate and clarified by centrifugation at $10,000 \times g$. The supernatant was purified through a 15–45% sucrose gradient at $151,000 \times g$ for 3 h at 4 °C. Gradient harvest was in 1 ml fractions from top to bottom followed by analysis for the presence of VLPs through immunoblotting and ELISA (Table 1 and Supplementary Fig. 3).

### Expression and purification of rsVLPs from plants (*Nicotiana benthamiana*)

The construction of pEAQ-*HT*-PV3 SktSC8-P1 and pEAQ-3CD, containing plant codon-optimised versions of the P1 region of PV3(Saukett)-SC8 and the PV 3CD protease from PV1 Mah have been reported previously[38]. Plasmids pEAQ-*HT*-PV1 MahSC6b-P1 and pEAQ-*HT*-PV2 MEFSC6b-P1 were constructed by inserting codon-optimised versions of the P1 region of either PV1 (Mahoney)-SC6b or PV2 (MEF-1)-SC6b between the *Age*I and *Xho*I sites of pEAQ-*HT*[66]. *Agrobacterium tumefaciens* strain LBA4404 was transformed with each construct separately by electroporation and propagated at 28 °C in Luria-Bertani media containing 50 μg/mL kanamycin and 50 μg/mL rifampicin. Co-expression of the P1 sequences with the 3CD protease in *Nicotiana benthamiana* leaves was carried out by co-infiltration as previously described[38] (Supplementary Fig. 4).

At various days post-infiltration (dpi) VLPs were extracted from the infiltrated region of the leaves with Dulbecco's phosphate-buffered saline + $MgCl_2$ and $CaCl_2$ (Sigma-Aldrich), in each case supplemented with EDTA to a final concentration of 20 mM. All extraction buffers also contained cOmplete protease inhibitor cocktail (Roche, UK). The purification process was carried out as previously described[38]. Briefly, crude extracts were centrifuged at $9500 \times g$ for 15 min at 4 °C following filtration over a 0.45 μm syringe filter (Sartorius). The clarified extract was then concentrated through a sucrose cushion (1 ml 70% (w/v) and 5 ml 25% (w/v)) at $167,000 \times g$ for 3 h at 4 °C and the lower fraction retrieved. Following dialysis and further concentration using PD10 desalting columns (GE Healthcare) and Amicon Centrifugal Filter Units (Millipore), the sample was purified by ultracentrifugation through a Nycodenz (Axis-Shield) gradient (20–60% (w/v)) at $247,103 \times g$ for 24 h and 4 °C. VLPs were collected by piercing the side of the tube with a needle and the Nycodenz removed through PD10 desalting columns (GE Healthcare) prior to concentration using Amicon Centrifugal Filter Units (Millipore) (Table 1 and Supplementary Fig. 4).

### Enzyme-linked immunosorbent assay (ELISA)

A non-competitive sandwich ELISA was used to measure the PV D Ag or C Ag[67]. Briefly, two-fold dilutions of antigen were captured with a serotype-specific polyclonal antibody, then detected using serotype-specific, D Ag- or C Ag-specific monoclonal antibodies followed by anti-mouse peroxidase conjugate. The D Ag content of each test sample was evaluated against a reference of assigned D Ag content[65] by parallel line analysis (Combistats). For D Ag specific ELISA the monoclonal antibodies used were 234 for type 1, 1050 for type 2 and 520 for type 3, and for C Ag specific ELISA 1588 for type 1 and 517 for type 3. No C Ag specific type 2 antibody was available.

## Thermostability assays

Thermostability of PV rsVLPs was assessed as in previous studies[31]. Briefly, the samples were diluted in DPBS to twice the concentration required to obtain an OD of 1.0 in the D Ag ELISA. Duplicate samples were heated for 10 min at a range of temperatures from 30 to 60 °C then diluted 1:1 with 4% dried milk in DPBS and cooled on ice. D Ag and C Ag content was measured by ELISA. The temperature at which the change from D Ag to C Ag occurred is recorded at the point where native antigenicity is reduced by 50%.

## Immunogenicity in rats

Immunogenicity of rsVLP preparations was assessed using pharmacopeial methods established at MHRA for the release of IPV lots. D Ag content was measured by ELISA and immunogenicity was assessed in female Wistar rats between 6 and 12 weeks old[57]. Groups of 10 rats per dose were immunised i.m. with 0.25 ml in each hind leg and terminal bleed collected on day 21. Sera were analysed for neutralising antibody response. The neutralising antibody responses to a range of antigen doses were compared to those elicited by a concurrently tested International Standard preparation.

Immunogenicity of rsVLP preparations formulated with adjuvant was assessed in the same way. Prior to inoculation of rats, samples were mixed with a 1/10th volume of Alhydrogel (2%, InvivoGen) and agitated for 30 min. At this time 100% of the D Ag was adsorbed onto the aluminium hydroxide (Supplementary Fig. 7).

## Live virus challenge of immunised TgPVR mice

TgPVR mice of both sexes (8 per test group aged 6–8 weeks) received one or two intraperitoneal or intramuscular injections of 0.2 ml PBS (controls) or the equivalent of 0.5 human doses of purified rsVLPs or the IPV European reference BRP[65]. The second dose, where given, was on day 14. Tail bleeds was taken prior to immunisation and challenge and sera were analysed for neutralising antibody response. Mice were challenged intramuscularly with 0.05 ml of 25 times the $PD_{50}$ of the relevant serotype of wt PV (Mahoney, MEF-1 or Saukett) then monitored for any signs of paralysis for 14 days[68]. Protection against challenge was compared with that in mice inoculated with PBS and with equivalent doses of the IPV reference preparation.

All animal experiments were performed under licences granted by the UK Home Office under the Animal (Scientific Procedures) Act 1986 revised 2013 and reviewed by the internal NIBSC Animal Welfare and Ethics Review Board. The TgPVR mouse and rat immunogenicity experiments were performed under Home Office licences PPL 70/8979, PPL 80/2478, PPL 80/2050, PPL 80/2537, P30D4C513, PP6108158, P856F6831 and P4F343A03.

## CryoEM sample preparation and data collection

Sucrose gradient purified fractions of each PV1-SC6b and PV2-SC6b rsVLP preparation (Fig. 2a) were buffer exchanged into DPBS-EDTA using Zeba Spin Desalting Columns (Thermo Fisher Scientific) with a 7 K molecular weight cut-off (MWCO) and in some cases further concentrated using Amicon Ultra Centrifugal Filter Units devices (100 kDa MWCO, Merck Millipore) to sample concentrations of ~0.1–1.1 mg/ml. For the PV1-SC6b$^{GPP3+GSH}$ complex, GPP3 was mixed at a ratio of 1 VLP:300 GPP3 molecules and incubated overnight at 4 °C, after which GSH was added to 10 mM final concentration and incubated on ice for 1–2 h. Compound GPP3 was dissolved in DMSO at 10 mg/ml and diluted to appropriate working stocks as required. GSH stock solutions were prepared in distilled $H_2O$ at pH 7.0.

CryoEM grid preparation was similar for all samples. Three to four microliters of rsVLP or rsVLP-compound-GSH mixture were applied to either glow-discharged C-flat™ holey carbon copper grids (product No. CF312, Electron Microscopy Sciences) for the PV1-SC6b yeast sample or Lacey carbon copper grids with an ultra-thin carbon support film (product No. AGS187-4, Agar Scientific) for all other samples. PV1-

SC6b yeast sample grids were prepared by double blotting to increase the number of particles in the holes. Briefly, after 30–60 s unbound sample was removed by manual blotting with filter paper, and grids were re-incubated with a further 3–4 μl of sample for 30 s, followed by mechanical blotting and rapid vitrification in a liquid nitrogen-cooled ethane/propane slurry with a Vitrobot Mark IV plunge-freezing device (Thermo Fisher Scientific) operated at 4 °C and 95–100% relative humidity using a blot force of −15 and blot time of 3.5 s. For all other samples, a single blotting procedure was performed on a Vitrobot Mark IV using the parameters above.

For PV1-SC6b (yeast and MVA) and PV1-SC6b$^{GPP3+GSH}$ (yeast) cryoEM data acquisition was performed at 300 kV with a Titan Krios G3i microscope (Thermo Fisher Scientific) equipped either with a Gatan K2 Summit direct electron detector (DED) with a Gatan GIF Quantum energy filter (PV1-SC6b yeast) or a Falcon III DED (Thermo Fisher Scientific) (PV1-SC6b MVA and PV1-SC6b$^{GPP3+GSH}$ yeast) at the OPIC electron microscopy facility, UK. Micrographs were collected as movies using a defocus range of -2.9 μm to -0.8 μm in either single-electron counting mode (PV1-SC6b yeast) or linear mode (PV1-SC6b MVA and PV1-SC6b$^{GPP3+GSH}$ yeast). Data were collected with a physical pixel size of either 1.05 Å per pixel (PV1-SC6b yeast) or 1.08 Å per pixel (PV1-SC6b$^{GPP3+GSH}$ yeast and PV1-SC6b MVA) resulting in calibrated magnifications of ×47,619 and ×129,629, respectively. This large discrepancy arises because of the very different pixel sizes of the K2 and Falcon III detectors used. CryoEM data for the PV2-SC6b VLPs (MVA and baculovirus produced) were collected at 300 kV on a Titan Krios (Thermo Fisher Scientific) equipped with either a K2 (PV2-SC6b MVA) or K3 (PV2-SC6b baculovirus) DED (Gatan) and GIF Quantum energy filters (Gatan) at the electron Bio-Imaging Centre, Diamond Light Source, UK. For PV2-SC6b (MVA) movies were collected in single-electron counting mode with a physical pixel size of 1.055 Å per pixel. For PV2-SC6b (baculovirus) the K3 DED was operated in super-resolution mode with a physical pixel size of 1.06 Å per pixel (0.53 Å per super-resolution pixel). For all samples data were typically collected with a total dose of -33-42 e⁻/Å² fractionated between 25-50 frames. Detailed sample-specific data acquisition parameters are summarised in Supplementary Table 1.

## CryoEM image processing

For PV1-SC6b (yeast) and PV1-SC6b (MVA) image processing and single-particle reconstruction were performed using RELION-3[69], while PV2-SC6b (baculovirus) used RELION-3.1[70]. Individual movie frames were aligned and averaged with dose weighting using MotionCor2[71] to produce images compensated for electron beam-induced specimen drift. Contrast transfer function (CTF) parameters were estimated using CTFFIND4[72]. Micrographs showing astigmatism, significant drift or crystalline ice rings were discarded. For PV1-SC6b (yeast) particle picking was performed using programme Xmipp[73] within the Scipion software framework[74], after which saved particle coordinates were imported into RELION. For PV1-SC6b (MVA) and PV2-SC6b (baculovirus) particle picking was performed using crYOLO[75] by first training the neural network on a randomly selected subset of -100 manually picked particles from micrographs covering a range of defocus values. Once trained crYOLO was used to pick the complete dataset in an automated manner, and the saved particle coordinates were then imported into RELION.

Single-particle structure determination for PV1-SC6b (yeast and MVA) and PV2-SC6b (baculovirus) used established protocols in RELION for image classification and gold-standard refinement to prevent over-fitting[76]. Particles (numbers given in Supplementary Table 1) were subjected to multiple rounds (at least two) of reference-free two-dimensional classification to discard bad particles and remove junk classes. The particle population for each dataset was further enriched by three-dimensional (3D) classification to remove broken and overlapping particles and to separate alternate conformations of VLPs (PV1-

SC6b yeast D Ag and C Ag VLPs). The initial model for each dataset was generated from the previously determined cryoEM structure of PV3-SC8 from a plant expression system[38] (EMDB accession code EMD-3747) low-pass filtered to 60 Å to avoid bias.

A final set of particles (Supplementary Table 1) were selected from the best-aligned 3D class averages for high-resolution 3D auto-refinement, with the application of icosahedral symmetry throughout. For each dataset, a representative class from the end of 3D classification was low pass filtered to 40 Å to avoid bias and used as a reference during refinement. After the first round of refinement the PV1-SC6b (yeast), PV1-SC6b (MVA) and PV2-SC6b (baculovirus) datasets were subjected to CTF refinement to estimate beam tilt, anisotropic magnification, per-particle defocus and astigmatism, and Bayesian polishing of beam-induced motion-correction with trained parameters[70]. This procedure was performed iteratively at least twice with 3D auto-refinement after each round. The final resolution was estimated using a Fourier shell correlation (FSC) threshold of 0.143[76] (Supplementary Fig. 5). The maps for each reconstruction were sharpened using Post-processing in RELION by applying automatically estimated inverse B-factors (Supplementary Table 1). Local resolution was estimated for each reconstruction using the RELION implementation of local resolution algorithm[69], and locally scaled maps were used for model building and refinement in all cases.

Data processing and single-particle reconstruction for both the PV1-SC6b[GPP3+GSH] complex (yeast) and PV2-SC6b rsVLP (MVA) was performed using CryoSPARC v4.2.1[77], following standard procedures for icosahedral reconstruction. Raw movies were aligned with Patch Motion Correction and CTF parameters estimated using Patch-CTF. Poor-quality images exhibiting significant drift, astigmatism or ice rings were discarded using the manual curation tool in CryoSPARC. Particles were initially blob-picked from a subset of images and subjected to a first 2D classification job to generate suitable templates, which were subsequently used to complete particle picking on the whole data set. Two-dimensional classification was performed iteratively at least twice to clean out junk particles, followed by the generation of five ab initio models with the application of icosahedral symmetry. Heterogeneous refinement with icosahedral symmetry was used to further refine the best-aligned particle sets to a single good-looking class. These particles were then subjected to homogeneous refinement with icosahedral symmetry and combined with CTF refinement and higher-order aberration correction. Final resolution was estimated using the gold-standard FSC 0.143 cut-off on maps output after automatic sharpening and local resolution estimation. Data processing statistics are summarised in Supplementary Table 1.

### Atomic model building, refinement and analysis

For all PV1-SC6b and PV2-SC6b reconstructions the atomic coordinates of the X-ray structure of PV1 (PDB 1HXS) or PV2 (PDB 1EAH), respectively were manually placed into the cryoEM electron potential maps using UCSF Chimera[78]. Manual fitting was optimised with the UCSF Chimera 'Fit in Map' command[78] and the 'Rigid Body Fit Molecule' function in Coot[79]. For the PV1-SC6b C Ag expanded conformation structures from yeast and MVA the fit of the initial model was further optimised with automatic molecular dynamics flexible fitting using Namdinator[80]. For all structures, the cryoEM map surrounding six neighbouring capsid protomers (each composed of subunits VP0, VP1 and VP3) was extracted using phenix.map_box within Phenix[81]. Manual rebuilding was performed on the central protomer model using the tools in Coot[79] and non-crystallographic symmetry operators were used to generate neighbouring protomers, followed by iterative positional and B-factor refinement in real-space using phenix.real_space_refine[82] within Phenix[81] to ensure stable refinement of protomer interfaces and clashes. All refinement steps were performed in the presence of hydrogen atoms. Chemical restraints for GPP3 were generated using the grade server[83]. Only atomic coordinates were refined; the maps were kept constant. Each round of model optimisation was guided by cross-correlation between the map and the model. Final models were validated using MolProbity[84], EMRinger[85] and CaBLAM[86] integrated within Phenix[81]. Refinement statistics are shown in Supplementary Table 2.

Structural superpositions and RMSD calculations were performed using programme SHP[87] and the 'LSQ superpose' and 'SSM superpose' tools within Coot[88]. Molecular graphics were generated using Pymol[89] and UCSF ChimeraX[90].

### Reporting summary

Further information on research design is available in the Nature Portfolio Reporting Summary linked to this article.

## Data availability

The atomic coordinates for the cryoEM structures in this study have been submitted to the Protein Data Bank (https://www.rcsb.org/) under the following accession codes (PDB ID): PV1-SC6b yeast D Ag particle (9EYY), PV1-SC6b yeast C Ag particle (9EZ0), PV1-SC6b[GPP3+GSH] yeast (9F3Q), PV1-SC6b mammalian C Ag particle (9F0K), PV2-SC6b mammalian (9F59), PV2-SC6b insect (9F5P). The cryoEM electron potential maps have been deposited in the Electron Microscopy Data Bank (https://www.ebi.ac.uk/emdb/) under the following accession codes (EMD ID): PV1-SC6b yeast D Ag particle (EMD-50064), PV1-SC6b yeast C Ag particle (EMD-50066), PV1-SC6b[GPP3+GSH] yeast (EMD-50176), PV1-SC6b mammalian C Ag particle (EMD-50112), PV2-SC6b mammalian (EMD-50189), PV2-SC6b insect (EMD-50199). The source data underlying Figs. 4–6 and Supplementary Figs. 6 and 7 are provided with this paper.

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

## Acknowledgements

Computation used the Oxford Biomedical Research Computing (BMRC) facility, a joint development between the Wellcome Centre for Human Genetics and the Big Data Institute supported by Health Data Research UK and the NIHR Oxford Biomedical Research Centre. Financial support was provided by the Wellcome Trust Core Award Grant Number 203141/Z/16/Z and which supports H.M.E.D. The views expressed are those of the author(s) and not necessarily those of the NHS, the NIHR or the Department of Health. The OPIC electron microscopy facility was founded by a Wellcome Trust JIF award (060208/Z/00/Z) and is supported by a Wellcome Trust equipment grant (093305/Z/10/Z). We acknowledge Diamond for access and support of the cryo-EM facilities at the UK National Electron Bio-Imaging Centre (eBIC), proposals EM14856-38 and EM20223-61. We are grateful for technical assistance from the OPIC and eBIC staff. M.W.B., C.P., L.D.C. and H.F. are supported by a WHO/Bill and Melinda Gates Foundation award (RG.IMCB.I8-TSA-083) and D.I.S. and E.E.F. are supported by the UK Medical Research Council (MR/N00065X/1) and E.E.F by the Wellcome Trust (101122/Z/13/Z). MVA-T7 virus was provided by Dr Houssam Attoui (formerly of The Pirbright Institute). The pMVA-GFP2 (p434) plasmid was kindly provided by Professor Sarah Gilbert and we thank A.V. Turner (formerly of The Jenner Institute, Oxford) for assistance with vector production. At JIC, the work was funded by the UK Biotechnological and Biological Sciences Research Council (BBSRC) Institute Strategic Programme Grants "Understanding and Exploiting Plant and Microbial Secondary Metabolism" (BB/J004596/1; GPL), "Molecules from Nature—Enhanced Research Capacity" (BBS/E/J/000PR9794; GPL), "Harnessing Biosynthesis for Sustainable Food and Health" (BB/X01097X/1; GPL) and the John

Innes Foundation (GPL). We also acknowledge the support of the horti-cultural services and bioimaging platforms at JIC (GPL). This work was performed as part of a collaborative effort funded by the WHO/Bill and Melinda Gates Foundation (RG.IMCB.I8-TSA-083) involving the follow-ing Institutions and individuals: University of Leeds: D.J. Rowlands, N.J. Stonehouse, L. Sherry, K. Grehan, J.J. Swanson, S. Matthews, C. Nicol. University of Oxford: D.I. Stuart, E.E. Fry, M.W. Bahar, C. Porta, L.de Colibus. Medicines and Healthcare Products Regulatory Agency (for-merly, National Institute for Biological Standards and Control): A.J. Macadam, P. Minor, H. Fox, S. Carlyle. John Innes Centre: G. P. Lomo-nossoff, D. Ponndorf, J. Marsian, I. Murdoch, D. Ponndorf, S-R. Kim, S. Shah. University of Florida: J.B. Flanegan. University of Reading: I.M. Jones, M. Uchida. The authors wish to thank members of our Scientific Advisory board (Jim Hogle, Ellie Ehrenfeld, Phil Minor, Jeff Almond, and Martin Eisenhawer) and members of the Pirbright Laboratory (Toby Tuthill and Joseph Newman) for their support and invaluable scien-tific input.

## Author contributions

Experiments were conceived and designed by D.J.R., N.J.S., G.P.L., A.J.M., E.E.F. and D.I.S. L.S., M.W.B., C.P., H.F., K.G., J.M., I.M., D.P., V.N., H.M.E.D., L.D.C., S-R.K., S.S., S.C., J.J.S., S.M. and C.N. performed experiments. M.W.B., H.M.E.D. and V.N. collected the cryoEM data. M.W.B. and V.N. processed the cryoEM data, built and refined atomic models and along with C.P., E.E.F. and D.I.S. analysed cryoEM results. M.W.B. and C.P. performed the mammalian and insect cell expression studies. L.S., K.G., J.J.S., S.M. and C.N. performed the yeast cell expression studies. J.M., I.M., D.P., S-R.K. and S.S. performed the plant expression studies. H.F. and S.C. performed the ELISA and thermo-stability assays and immunogenicity and live virus challenge studies, and H.F., S.C. and A.J.M. analysed the results. All authors interpreted the results. L.S., M.W.B., C.P., E.E.F., D.J.R., N.J.S., G.P.L., A.J.M. and D.I.S. wrote the manuscript, and all authors reviewed and edited the manuscript.

## Competing interests

A.J.M. declares he is the named inventor on International Patent Appli-cation No. PCT/GB2018/050129 (WO2018134584A1) and granted patent US20190358315A1 which covers the mutations used to stabilise the VLPs. G.P.L. declares that he is a named inventor on granted patent WO 29087391 A1 which describes the pEAQ vector system used for the plant expression studies in this manuscript. The remaining authors declare no conflict of interest.
