## [Peer Review file · Nature Communications]

Recombinant expression systems for production of stabilised virus-like particles as next-generation polio vaccines

Corresponding Author: Professor David Rowlands

Version 1:

Reviewer comments:

Reviewer #1

(Remarks to the Author)

The authors attempted to develop new poliovirus vaccines using highly stable virus-like particles (VLPs). They produced VLPs from all three serotypes using four recombinant protein expression systems. The immunogenicity of these VLPs was evaluated, and structural analysis was performed. These results are important for the development of novel vaccines; however, numerous concerns exist regarding the experimental data and design of the vaccination studies.

Please describe how many micrograms of inactivated poliovirus vaccine (IPV) or VLP were administered to mice and rats.

Figure 4: Did the authors compare the vaccine efficacy between wild-type VLP and stabilized VLP? I could not properly determine the usefulness of stabilized VLPs in vaccines.

Figure 4: Why did the authors use intraperitoneal rather than muscle administration? This difference in the route of administration may have resulted in the conflicting results between mice and rats. The authors should examine the effects of the vaccine on muscle vaccination in mice.

Figure 4: No antibody titers were detected in mammalian-derived type 1 and plant-derived type 2 VLPs. Therefore, an explanation of the structural information is required.

Figure 4: There are several bars in the IPV group. What exactly do these mean?

Figure 6: Contrary to the results in mice, VLPs had lower antibody titers compared to IPV for types 1 and 2. Considering that the rat study was performed using muscular vaccination, its results may be more extrapolative to humans than those of the mouse study. Given these results, whether VLPs are superior to IPV remains unclear. Therefore, the authors should investigate the effects of this vaccine using muscular vaccination in mice.

Figure 6b: For type 1, the antibody titer increased in yeast-derived VLPs when combined with alum but not in insect-derived VLPs. Why is this?

Figure 6b: I speculate that IPV also increased antibody titers when used in combination with alum. Therefore, VLPs are inferior to IPV in mice.

Reviewer #2

(Remarks to the Author)

This is an important study of stabilized VLPs with vaccine potential produced from four expression systems. The investigators demonstrated equal or better antigenicity, thermostability, and immunogenicity of these stabilized VLPs

compared to the current inactivated polio vaccine, and solved the 3D structures to understand the contributions of stabilizing mutations. The poliovirus stabilized VLPs are important players in development of viable next-generation vaccine candidates for the future. There are no major concerns and only minor comments for this well written manuscript. For the VLPs from yeast, if the D and C antigens were separated during the cryoEM reconstruction process, what was the ratio found to be during classification?

The cryoEM structures are nicely resolved and presented well. However, it seems likely that local resolution mapping would complement the work by illustrating the enhanced stability of the VLPs. Similarly, details of the stabilizing effect of rsVLPs by GPP3 and GSH might be illustrated with local resolution mapping.

Reviewer #3

(Remarks to the Author)

The authors have presented an impressive paper exploring most major eukaryotic expression systems to produce stable poliovirus virus-like particles (VLPs) for all three serotypes. These recombinant stabilized VLPs (rsVLPs) were rigorously characterized for their thermal stability and antigenic properties, demonstrating equivalent or superior immunogenicity compared to the current inactivated poliovirus vaccine (IPV). Since these VLPs are produced using DNA expression vectors and lack the poliovirus genomic RNA inside, they offer a significantly safer alternative to the current oral poliovirus vaccine (OPV). The reviewer was pleasantly surprised to see such extensive and thorough research presented as a communication-type publication. Nevertheless, the reviewer commends the work, recognizing its strong potential for rsVLPs to replace current IPV and OPV as a much safer vaccine candidate. Additionally, these findings pave the way for the application of similar VLP vaccines to other emerging enteroviruses. The reviewer has only a few suggestions to further improve the quality of the paper.

1. Pg16Ln357 to Ln369 in the discussion section appears repetitive of Pg3Ln57 to Pg4Ln73 in the introduction section. Both sections describe the necessity and significance of using rsVLPs as vaccines compared to the current vaccines. In the reviewer's opinion, this information is more appropriate for the introduction. The discussion should focus more on the results rather than the purpose of the research.

2. Pg11Ln249 "cryoEM structure analyses of PV rsVLPs". First, consider changing "analysis" to "analyses" in the subsection title. The reviewer suggests that the authors should weaken or at least discuss with caution the D Ag or C Ag in the reconstructions, as cryoEM classification and averaging might introduce bias into the final structure. To thoroughly address this, the authors need to carefully perform multiple-model reconstructions with more statistical analyses. Otherwise, the final reconstructed map might represent just one of the dominant homogeneous species in the sample. Additionally, inconsistencies between the D and C Ag in the reconstruction and ELISA or other biochemical results might be introduced during cryoEM sample preparation, such as buffer exchange, storage, and temperature changes.

3. Pg17Ln399 to Pg18Ln409 discusses the stability differences among rsVLPs expressed by different systems. The authors explored the potential role of the pocket factor. However, the VLPs might also carry some RNA or nucleic acid oligomers from the cell that stabilize the VLPs. The internal surfaces of enteroviruses have positively charged residues and aromatic residues at certain places. The positively charged residues can bind to the phosphate backbone of nucleic acids, and the hydrophobic aromatic side chains can stack with nucleic acid bases. These nucleic acids, "stolen" from different expression systems, might stabilize the VLPs differently. Have the authors tested whether their rsVLPs contain nucleic acid inside? The authors mentioned on Pg19Ln440 that the viral RNA from IPV might act as an adjuvant. It is not clear to the reviewer whether the expressed rsVLPs might contain nucleic acids inside.

4. Pg20Ln453 to Pg22Ln559 in the method section described the expression and purification of rsVLPs from different systems. The authors should cite their supplementary figures, which contain valuable cloning construction details and VLP purification flow charts. However, none of these figures have been cited in the manuscript. The reviewer searched the entire manuscript and found citations only for supplementary tables.

5. Pg24Ln558: "The purification process was carried out at 4°C." Compared to the corresponding purification protocols in other systems, this section lacks the necessary details for other researchers to reproduce the results.

6. g25Ln585: An extra closing parenthesis should be removed from the end of the paragraph.

7. Pg25Ln590: "At this time, 100% of the D Ag was adsorbed onto the aluminium hydroxide (not shown)." This statement lacks an explanation and justification for the "not shown" data.

8. Pg27Ln631 to Ln633: The two small pixel sizes of 1.05Å and 1.08Å correspond to large magnification differences of x47,619 and x129,629, respectively (approximately three times). This discrepancy needs some explanation or clarification.

9. Pg28Ln687 to Ln688: "Astigmatism or ice rings were discarded using the manual curation tool." Some additional description is needed here to ensure reproducibility. Are these tools available? At the very least, general criteria for the discarding need to be established and described.

10. Pg31Ln732: "The source data underlying Fig. X are provided in Supplementary Data X." It appears that the authors might have forgotten to update this information in their manuscript.

11. Curiosity from the reviewer: Why is the purification of VLP from the yeast system done at 10°C, while other systems use 4°C?

12. Pg44Ln1046: Figure 2 subpanel C. It is very hard to see the colored residues on the whole virus surface. The reviewer suggests showing a zoomed-in version that only displays the asymmetric unit instead of the entire icosahedral virus.

13. Pg46Ln1063: Figure 3 subpanels a and c. The reconstructed VLPs are presented using a color scheme based on the radius to the virus center, but the color ranges vary. The minimum values range from 105 to 115, and the maximum values range from 160 to 175. The reviewer suggests that the authors use the same color key for better comparison. Additionally, why is subpanel e colored by VPs instead of the radius? The reviewer suggests using the same coloring scheme as the other subpanels.

14. Supplementary materials: For readability purposes, the reviewer suggests moving Table S1 and S2 after Figure S4 so that all the cryo-EM data are together. However, this adjustment might need to be made in accordance with the journal's

requirements. As stated above, please cite all the supplementary figures in main manuscript.

15. Supplementary materials: The reviewer suggests making the word choice consistent with the citations in the main manuscript. Choose either "supplementary" or "supplemental" and use it consistently.

Version 2:

Reviewer comments:

Reviewer #1

(Remarks to the Author)

In the revised paper, the authors addressed all my questions. I recommend publication.

Reviewer #2

(Remarks to the Author)

Authors have addressed all issues adequately.

Reviewer #3

(Remarks to the Author)

The authors have addressed the reviewers' comments thoroughly, and the manuscript has improved. However, there are a few issues with the line numbers referenced in the authors' responses, which do not correspond to the marked-up version of the manuscript. This discrepancy makes the review process challenging, as the reviewer must rely on the original submission from the first round to locate the corresponding sections.

For instance:

Comment #1: Lines 458–464 should correspond to lines 369–375 in the marked-up version.

Comment #2: Lines 333–335 should correspond to lines 260–262.

Comment #5: Lines 699–708 should correspond to lines 570–579.

Comment #8: Lines 793–795 should correspond to lines 653–655.

Comment #9: Line 858 should correspond to line 710.

For Comment #3, the observation that VLPs do not package RNA from the expression hosts is interesting. Adding a brief discussion, even a single sentence with citation, would enhance the manuscript.

Regarding Comment #7, in the marked-up version, line 610 references "Supplementary Fig. 6," but in the final version, line 606 correctly references "Supplementary Fig. 7." Please ensure that the final published version reflects Fig. 7.

Finally, a minor editorial suggestion: In the caption for Fig 2 in the final version, the last sentence reads, "The panels beneath each capsid representation show an expanded view of each AU." Consider changing "Enlarged views of each AU were presented beneath the corresponding capsids" for improved clarity and precision.

Response to the Reviewers

We are very grateful to the reviewers for their helpful comments, addressed below. We feel that the manuscript is much improved as a result. All line references refer to the marked-up version of the manuscript.

Reviewer Comments

Reviewer #1 (Remarks to the Author):

The authors attempted to develop new poliovirus vaccines using highly stable virus-like particles (VLPs). They produced VLPs from all three serotypes using four recombinant protein expression systems. The immunogenicity of these VLPs was evaluated, and structural analysis was performed. These results are important for the development of novel vaccines; however, numerous concerns exist regarding the experimental data and design of the vaccination studies.

Please describe how many micrograms of inactivated poliovirus vaccine (IPV) or VLP were administered to mice and rats.

- IPV is traditionally measured in D antigen units, as highlighted by the current vaccine dose requirements, which are 32, 8, and 28 D antigen units for serotype 1, 2, and 3 respectively. Therefore, in order to ascertain the immunogenicity of our VLPs in direct comparison to IPV, we determined the amount of D antigen for each serotype to match the IPV inoculum.

As this is the standard unit of measurement for all poliovirus vaccines, we do not believe it is helpful to amend the manuscript to describe the amount of IPV or VLP in micrograms.

Figure 4: Did the authors compare the vaccine efficacy between wild-type VLP and stabilized VLP? I could not properly determine the usefulness of stabilized VLPs in vaccines.

- It has been well established that wild-type empty capsids or VLPs are highly unstable, leading to a conformational change from the native D antigen to C antigen, an expanded conformation which does not induce a protective neutralising response as outlined in Lines 101-106. Therefore, in line with the NC3Rs, we chose to reduce the number of animal groups tested as it had already been established that wt VLPs would be ineffective as vaccine candidates (Rombaut & Jore, 1997, reference 41 in the manuscript).

Figure 4: Why did the authors use intraperitoneal rather than muscle administration? This difference in the route of administration may have resulted in the conflicting results between mice and rats. The authors should examine the effects of the vaccine on muscle vaccination in mice.

- We opted for intraperitoneal administration in our mice experiments as this is the standardised protocol, described in Martin et al, 2003 (Ref. 74 in the manuscript),

which allows the comparison of these experiments with previously published immunisation assays using PVR transgenic mice.

In addition, during our investigation we have previously compared intraperitoneal and intramuscular administration in our mouse model using rsVLPs from insect and mammalian cells and observed no significant difference in seroconversion, these data are now shown in new Supplementary Fig. 6 and referred to in the manuscript in Lines 445-449.

Therefore, as no discernible differences are seen using alternative administration routes, we do not believe it would be ethical to repeat all the experiments in mice using intramuscular administration.

Figure 4: No antibody titers were detected in mammalian-derived type 1 and plant-derived type 2 VLPs. Therefore, an explanation of the structural information is required.

- For mammalian type 1 rsVLP, we only reconstructed a C Ag particle, although there was transient DA_g detected for some preps of PV1-SC6b. Because of the low amount of DA_g particles the experiment in Figure 4 was not performed – the figure has been modified to reflect this.

We did not obtain sufficient type 2 VLPs from plants to carry out structural analysis or to immunise mice.

Figure 4: There are several bars in the IPV group. What exactly do these mean?

- The IPV bars show the positive control response in comparison to the parallel VLP response in each experiment. The colour of the stripes in each IPV bar indicates for which expression system(s) IPV was the positive control. The figure legend has been modified to clarify this for the reader.

Figure 6: Contrary to the results in mice, VLPs had lower antibody titers compared to IPV for types 1 and 2. Considering that the rat study was performed using muscular vaccination, its results may be more extrapolative to humans than those of the mouse study. Given these results, whether VLPs are superior to IPV remains unclear. Therefore, the authors should investigate the effects of this vaccine using muscular vaccination in mice.

- We agree that the rat study is more extrapolative to humans as this is used as the ‘gold standard’ lot batch release assay for poliovirus vaccines. The rat model shows a good dose-response curve following immunisation, mirroring the dose-response curve to that seen in humans (Van Steenis et al., 1980, Ref. 58 in the manuscript).

However, there is no difference in neutralising antibody responses to IPV following intramuscular administration compared to intraperitoneal administration in the mouse model, as now highlighted in the new Supplementary Fig. 6. Therefore, we do not believe that further mouse experiments are required to investigate this.

Figure 6b: For type 1, the antibody titer increased in yeast-derived VLPs when combined with alum but not in insect-derived VLPs. Why is this?

- We agree that in the presence of adjuvant, type 1 insect-derived VLPs do not show such a large increase in neutralising antibody titre as the yeast-derived VLPs at a single human dose. Nevertheless, there is a significant improvement, and below a single human dose they slightly outperform IPV.

Furthermore, as these VLPs are produced from different expression systems, it is possible that some contaminating host proteins may influence the neutralising antibody response. If these proteins are immunosuppressive, this would be more prevalent in animal groups which received the largest immunisation dose. This may explain the comparatively reduced improvement in neutralising antibody titre following administration of a full human dose.

Figure 6b: I speculate that IPV also increased antibody titers when used in combination with alum. Therefore, VLPs are inferior to IPV in mice.

- The aim of this experiment was to show that VLPs could be as immunogenic as IPV in rats when administered with an adjuvant which is currently used in other licensed VLP vaccines.

We agree that it would be interesting to determine whether adjuvants also boost the neutralising antibody response to IPV and are investigating this as part of a separate project. These data will be included in a different manuscript that is currently in preparation. However, we believe that our major conclusion that adjuvanted VLPs display equal or superior immunogenicity in comparison to unadjuvanted IPV is valid.

Reviewer #2 (Remarks to the Author):

This is an important study of stabilized VLPs with vaccine potential produced from four expression systems. The investigators demonstrated equal or better antigenicity, thermostability, and immunogenicity of these stabilized VLPs compared to the current inactivated polio vaccine, and solved the 3D structures to understand the contributions of stabilizing mutations. The poliovirus stabilized VLPs are important players in development of viable next-generation vaccine candidates for the future. There are no major concerns and only minor comments for this well written manuscript.

For the VLPs from yeast, if the D and C antigens were separated during the cryoEM reconstruction process, what was the ratio found to be during classification?

- A ratio of approximately 1:1.8 between DAg and CAg particles for the PV1-SC6b rsVLP from yeast. The results section has been modified to describe this, lines 340-341.

The cryoEM structures are nicely resolved and presented well. However, it seems likely that local resolution mapping would complement the work by illustrating the enhanced stability of the VLPs. Similarly, details of the stabilizing effect of rsVLPs by GPP3 and GSH might be illustrated with local resolution mapping.

- Although we do not have cryoEM reconstructions of the unstabilised version of the VLPs (i.e. wild-type VLP), we can show the differences in local resolution mapping between PV1-SC6b (Supplementary Fig. 5a) and PV1-SC6b+GPP3+GSH (Supplementary Fig. 5e).

Reviewer #3 (Remarks to the Author):

The authors have presented an impressive paper exploring most major eukaryotic expression systems to produce stable poliovirus virus-like particles (VLPs) for all three serotypes. These recombinant stabilized VLPs (rsVLPs) were rigorously characterized for their thermal stability and antigenic properties, demonstrating equivalent or superior immunogenicity compared to the current inactivated poliovirus vaccine (IPV). Since these VLPs are produced using DNA expression vectors and lack the poliovirus genomic RNA inside, they offer a significantly safer alternative to the current oral poliovirus vaccine (OPV). The reviewer was pleasantly surprised to see such extensive and thorough research presented as a communication-type publication. Nevertheless, the reviewer commends the work, recognizing its strong potential for rsVLPs to replace current IPV and OPV as a much safer vaccine candidate. Additionally, these findings pave the way for the application of similar VLP vaccines to other emerging enteroviruses. The reviewer has only a few suggestions to further improve the quality of the paper.

1. Pg16Ln357 to Ln369 in the discussion section appears repetitive of Pg3Ln57 to Pg4Ln73 in the introduction section. Both sections describe the necessity and significance of using rsVLPs as vaccines compared to the current vaccines. In the reviewer's opinion, this information is more appropriate for the introduction. The discussion should focus more on the results rather than the purpose of the research.

- We agree and have updated the text in these sections accordingly, removing lines 458-464 to focus the discussion towards the results.

2. Pg11Ln249 "cryoEM structure analyses of PV rsVLPs". First, consider changing "analysis" to "analyses" in the subsection title. The reviewer suggests that the authors should weaken or at least discuss with caution the D Ag or C Ag in the reconstructions, as cryoEM classification and averaging might introduce bias into the final structure. To thoroughly address this, the authors need to carefully perform multiple-model reconstructions with more statistical analyses. Otherwise, the final reconstructed map might represent just one of the dominant homogeneous species in the sample. Additionally, inconsistencies between the D and C Ag in the reconstruction and ELISA or other biochemical results might be introduced during cryoEM sample preparation, such as buffer exchange, storage, and temperature changes.

- The DAg and CAg structures are sufficiently different that the 3D classification can be considered robust, and this is reflected in the relatively good resolution of the DAg and CAg reconstructions.

We agree with the reviewers point that sample preparation may account for differences observed between the reconstructions and the ELISA assays – for example in the case of PV1-SC6b from mammalian expression (some DAg in ELISA, but only CAg by cryoEM). We have added a sentence to the results section to reflect this, lines 333 to 335.

3. Pg17Ln399 to Pg18Ln409 discusses the stability differences among rsVLPs expressed by different systems. The authors explored the potential role of the pocket factor. However, the VLPs might also carry some RNA or nucleic acid oligomers from the cell that stabilize the VLPs. The internal surfaces of enteroviruses have positively charged residues and aromatic residues at certain places. The positively charged residues can bind to the phosphate backbone of nucleic acids, and the hydrophobic aromatic side chains can stack with nucleic acid bases. These nucleic acids, "stolen" from different expression systems, might stabilize the VLPs differently. Have the authors tested whether their rsVLPs contain nucleic acid inside? The authors mentioned on Pg19Ln440 that the viral RNA from IPV might act as an adjuvant. It is not clear to the reviewer whether the expressed rsVLPs might contain nucleic acids inside.

- The OD260/280 ratio reflects the presence of nucleic acids and we did not detect RNA in the samples we tested. We have previously investigated the presence of RNA within *Pichia*-derived VLPs, initially using OD260/280, which provided no evidence for the presence of RNA (PMID: 32161150). We then used the highly sensitive Particle Stability Thermal Release assay (PaSTRy), which uses Syto9, a dye which fluoresces upon interaction with RNA. Whilst a strong signal for RNA was detected from infectious virus particles following thermal stressing, neither wt nor rsVLPs activated Syto9, suggesting the absence of encapsidated nucleic acid. (PMID: 36298714).

4. Pg20Ln453 to Pg22Ln559 in the method section described the expression and purification of rsVLPs from different systems. The authors should cite their supplementary figures, which contain valuable cloning construction details and VLP purification flow charts. However, none of these figures have been cited in the manuscript. The reviewer searched the entire manuscript and found citations only for supplementary tables.

- We have ensured we have referred to the relevant supplementary figures throughout the methods section.

5. Pg24Ln558: "The purification process was carried out at 4°C." Compared to the corresponding purification protocols in other systems, this section lacks the necessary details for other researchers to reproduce the results.

- This has been updated in the manuscript to provide more detail for the purification of plant-derived VLPs. Lines 699 to 708.

6. Pg25Ln585: An extra closing parenthesis should be removed from the end of the paragraph.

- This has been updated.

7. Pg25Ln590: "At this time, 100% of the D Ag was adsorbed onto the aluminium hydroxide (not shown)." This statement lacks an explanation and justification for the "not shown" data.

- We have amended the manuscript to provide representative data of D Ag absorption to the aluminium hydroxide adjuvant in the supplementary data. These data are now shown in new Supplementary Fig. 7.

8. Pg27Ln631 to Ln633: The two small pixel sizes of 1.05Å and 1.08Å correspond to large magnification differences of x47,619 and x129,629, respectively (approximately three times). This discrepancy needs some explanation or clarification.

- The reason for this discrepancy is because the direct electron detectors used had significantly different pixel dimensions (K2, 5µm; and Falcon III, 14µm). This explanation has been added to the text for clarity, Lines 793 to 795.

9. Pg28Ln687 to Ln688: "Astigmatism or ice rings were discarded using the manual curation tool." Some additional description is needed here to ensure reproducibility. Are these tools available? At the very least, general criteria for the discarding need to be established and described.

- The standard tool within the cryoSPARC software package was used, and text has been added to clarify this (Line 858).

10. Pg31Ln732: "The source data underlying Fig. X are provided in Supplementary Data X." It appears that the authors might have forgotten to update this information in their manuscript.

- This text has been updated.

11. Curiosity from the reviewer: Why is the purification of VLP from the yeast system done at 10°C, while other systems use 4°C?

- Once we had established that the rsVLPs were thermally stable, we adapted the purification to be done at 10°C for equipment maintenance, reducing the stress on the ultracentrifuges to maintain the samples at 4°C throughout the centrifugation.

12. Pg44Ln1046: Figure 2 subpanel C. It is very hard to see the colored residues on the whole virus surface. The reviewer suggests showing a zoomed-in version that only displays the asymmetric unit instead of the entire icosahedral virus.

- Figure 2c has been amended to display an asymmetric unit highlighting the coloured residues in addition to the entire icosahedral virus.

13. Pg46Ln1063: Figure 3 subpanels a and c. The reconstructed VLPs are presented using a color scheme based on the radius to the virus center, but the color ranges vary. The minimum values range from 105 to 115, and the maximum values range from 160 to 175. The reviewer suggests that the authors use the same color key for better comparison. Additionally, why is subpanel e colored by VPs instead of the radius? The reviewer suggests using the same coloring scheme as the other subpanels.

- The colouring has been updated to be on the same range in panels a, c and d. The colouring of panel (e) was chosen to better relate the bound GSH (magenta) to the capsid protein (VP1 – blue).

14. Supplementary materials: For readability purposes, the reviewer suggests moving Table S1 and S2 after Figure S4 so that all the cryo-EM data are together. However, this adjustment might need to be made in accordance with the journal's requirements. As stated above, please cite all the supplementary figures in main manuscript.

- This will be down to the journal style but all supplementary figures are now cited.

15. Supplementary materials: The reviewer suggests making the word choice consistent with the citations in the main manuscript. Choose either "supplementary" or "supplemental" and use it consistently.

- This has been checked relative to the journal style and updated as appropriate.

Response to the Reviewers

We are very grateful to the reviewers for their helpful comments, addressed below. We feel that the manuscript is improved as a result. All line references refer to the clean copy of the manuscript.

Reviewer Comments

Reviewer #1 (Remarks to the Author):

In the revised paper, the authors addressed all my questions. I recommend publication.

- We thank the reviewer for their time and effort to improve our manuscript.

Reviewer #2 (Remarks to the Author):

Authors have addressed all issues adequately.

- We thank the reviewer for their time and effort to improve our manuscript.

Reviewer #3 (Remarks to the Author):

The authors have addressed the reviewers' comments thoroughly, and the manuscript has improved. However, there are a few issues with the line numbers referenced in the authors' responses, which do not correspond to the marked-up version of the manuscript. This discrepancy makes the review process challenging, as the reviewer must rely on the original submission from the first round to locate the corresponding sections.

For instance:

Comment #1: Lines 458–464 should correspond to lines 369–375 in the marked-up version.

Comment #2: Lines 333–335 should correspond to lines 260–262.

Comment #5: Lines 699–708 should correspond to lines 570–579.

Comment #8: Lines 793–795 should correspond to lines 653–655.

Comment #9: Line 858 should correspond to line 710.

- We would like to thank the reviewer for their time and effort to improve our manuscript and apologise for the issue with the line numbers referenced, we appreciate the efforts of the reviewer to ensure our manuscript had been revised adequately.

For Comment #3, the observation that VLPs do not package RNA from the expression hosts is interesting. Adding a brief discussion, even a single sentence with citation, would enhance the manuscript.

- We agree that this is an interesting observation, and we have highlighted this alongside a citation describing this work in the Discussion, Line 447-449.

Regarding Comment #7, in the marked-up version, line 610 references "Supplementary Fig. 6," but in the final version, line 606 correctly references "Supplementary Fig. 7." Please ensure that the final published version reflects Fig. 7.

- We have ensured the final version correctly references Supplementary Fig. 7 in Line 606.

Finally, a minor editorial suggestion: In the caption for Fig 2 in the final version, the last sentence reads, "The panels beneath each capsid representation show an expanded view of each AU." Consider changing "Enlarged views of each AU were presented beneath the corresponding capsids" for improved clarity and precision.

- We thank the reviewer for this suggestion and have amended the legend as requested.